

# Improvement of Airborne Retrievals of Cloud Droplet Number Concentration of Trade Wind Cumulus Using a Synergetic Approach.

Kevin Wolf[1], André Ehrlich[1], Marek Jacob[2], Susanne Crewell[2], Martin Wirth[3], and Manfred Wendisch[1]

[1]Leipzig Institute for Meteorology, University of Leipzig, Leipzig, Germany

[2]Institute for Geophysics and Meteorology, University of Cologne, Cologne, Germany

[3]Institute of Atmospheric Physics, German Aerospace Center, Oberpfaffenhofen, Germany

*Correspondence to:* K. Wolf (kevin.wolf@uni-leipzig.de)

**Abstract.** The cloud droplet number concentration $N$ is a major variable in numerical weather prediction (NWP) and global climate models (GCMs). In combination with the liquid water content $LWC$ it determines the cloud droplet effective radius $r_{\mathrm{eff}}$, which is used to calculate the radiative effects of clouds. Especially, the solar cloud top reflectivity $\mathcal{R}$ of shallow trade wind cumulus is highly sensitive with respect to $N$ and $LWC$.

In-situ measurements of $N$ are limited by the sampled cloud volume, not covering the significant natural variability. Satellite retrievals of $N$ suffer from large uncertainties and strictly assume adiabatic vertical profiles of cloud properties. Therefore, it is suggested to use the synergy of passive and active airborne remote sensing to reduce the uncertainty of $N$ retrievals and to bridge the gap between in-situ cloud sampling and global averaging.

  For this purpose, spectral solar radiation measurements above shallow trade-wind cumulus were combined with passive mi-
crowave and active radar and lidar observations carried out during the second Next Generation Remote Sensing for Validation Studies (NARVAL-II) campaign with the High Altitude and Long Range Research Aircraft (HALO) in August 2016. The common approach to retrieve $N$ is further developed by including combined measurements and retrievals of cloud optical thickness $\tau$, liquid water path $LWP$, $r_{\mathrm{eff}}$, as well as cloud base and top altitude. Three approaches are tested and applied to synthetic measurements and two cloud cases of NARVAL-II. Using the new techniques, errors in $N$ due to the adiabatic assumption
have been reduced significantly.



# 1 Introduction

Clouds strongly influence the Earth's radiation budget by reflecting, absorbing, and emitting solar and terrestrial radiation. These effects are typically quantified by the cloud radiative forcing (CRF), which is defined by the difference between the net radiation (irradiance) in cloudy and cloud-free conditions. Depending on cloud type, their optical and microphysical properties, as well as their spatial and temporal occurrence, the CRF can vary significantly (Rosenfeld, 2006). In the tropics, clouds can either cool or warm the atmosphere / surface below the cloud. While for cirrus a warming effect dominates, boundary layer trade wind cumulus typically cool the subjacent atmosphere / surface by efficiently reflecting solar radiation (Warren et al., 1988). Therefore, a realistic representation of these clouds in numerical weather prediction (NWP) and global climate models (GCMs) is essential. Due to their sub-grid scale, internal variability, and boundary layer interactions, trade wind cumulus are not well represented in NWP and GCMs (Kollias and Albrecht, 2010). A major source of uncertainty of these models is caused by insufficient representation of the first aerosol effect (Bony and Dufresne, 2005), which describes the correlation of the cloud droplet number concentration $N$ and the cloud optical thickness $\tau$ or cloud top reflectivity $\mathcal{R}$, commonly known as the Twomey effect (Twomey, 1977). It is most effective for optically thin, low-level clouds such as trade wind cumulus (Platnick and Twomey, 1994; Werner et al., 2014), which are an ubiquitous cloud type in the tropics (Warren et al., 1988; Eastman et al., 2011). Despite their small vertical and horizontal extent they can have fractional cloudiness of more than $25\%$ (Albrecht, 1991) and, therefore, may influence the Earth radiation budget significantly (Chertock et al., 1993). Additional, trade-wind cumulus plays an important role in maintaining the thermodynamic budget in the atmospheric boundary layer. They couple the surface and free atmosphere by transporting latent heat and developing deep convection (Lamer et al., 2015). Another important factor determining the CRF is the number concentration of aerosol particles, in particular the amount of particles which can act as cloud condensation nuclei (CCN) (Werner et al., 2013). Depending on the CCN number concentration, precipitation formation can be promoted or inhibited (Lee and Feingold, 2013). The CCN concentration influences the cloud life cycle and life time (Albrecht, 1989). The magnitude of both effects depend on the individual cloud regime.

Operational NWP models usually do not have the computational capability to consider size-resolved microphysical schemes and, therefore, the usage of simplified parametrizations is inevitable. The most important parameter, which links microphysical and radiative properties of clouds, is the effective radius $r_{\mathrm{eff}}$, which represents the radiatively effective size of a particle population (Pontikis and Hicks, 1992). In NWP and GCMs, $r_{\mathrm{eff}}$ is calculated from the cloud droplet number concentration $N$ and the liquid water content $LWC$. In simple models, assumptions of constant $N$ are applied for different situations, e.g. the separation of polluted and unpolluted air-masses, which is unrealistic. As $r_{\mathrm{eff}}$ is calculated from $LWC$ and $N$, the cloud droplet number concentration is a key parameter for models to calculate reasonable values of $r_{\mathrm{eff}}$ and to represent the Twomey effect.

The cloud optical thickness $\tau$ of a homogeneous cloud layer, is estimated from $LWC$ and $r_{\mathrm{eff}}$ by:

$$\tau = \frac{3 \cdot \int_{\mathrm{h_{base}}}^{\mathrm{h_{top}}} LWC \cdot \mathrm{d}z}{2 \cdot \rho_{\mathrm{w}} \cdot r_{\mathrm{eff}}}, \tag{1}$$



following Hansen and Travis (1974) and Stephens (1978), with $\rho_{\mathrm{w}}$ the density of liquid water, $h_{\mathrm{base}}$ the cloud base height, and $h_{\mathrm{top}}$ the cloud top height. Equation (1) assumes a homogeneous vertical cloud profile, which is not a realistic scenario for most of the observed clouds, as they are usually sub-adiabatic (Brenguier et al., 2000; Painemal and Zuidema, 2011; Min et al., 2012).

To measure $N$ and $LWC$, airborne in-situ measurements are applied, utilizing different physical methods and instruments (Baumgardner et al., 2011; Wendisch and Brenguier, 2013). These are based on optical measurement principles such as forward scattering, phase doppler interferometry and holographic imaging. Beside the uncertainties of the individual measurement techniques, the total sample volume of the instruments is rather limited in comparison to the typical horizontal and vertical extent of clouds. Due to the limited flight time and range, airborne in-situ observations cover the natural variability of $N$,

$r_{\mathrm{eff}}$, and $LWC$ partly. To directly quantify the Twomey effect, co-located measurements of cloud microphysical and radiative properties are required, which was realized only in a few occasions (Ackerman et al., 2000; Siebert et al., 2013; Werner et al., 2014).

Satellite remote sensing products are developed, for example by Quaas et al. (2009); Minnis et al. (2011); Mace et al. (2016); Bennartz and Rausch (2017). They are a useful tool, as they provide large spatial and temporal data sets. To improve global

statistics of estimates of the Twomey effect, several approaches to derive $N$ from satellite observations have been developed (Grosvenor et al., 2018b). Based on passive remote sensing in the solar and terrestrial wavelength range, $N$ is estimated combining the results of bi-spectral retrievals of $\tau$ and $r_{\mathrm{eff}}$ and cloud top temperature $T_{\mathrm{top}}$ by Brenguier et al. (2000), Quaas et al. (2006), Zeng et al. (2014), and Bennartz and Rausch (2017). They assume a constant $LWC$ and $N$ throughout the cloud vertical profile, which is not necessarily fulfilled in nature. More complex vertical profile types of $LWC$ and $N$ are applied by

Boers et al. (2006), where an inhomogeneous mixing model assumes that entrainment dilutes the air parcel with constant volume radius of the droplets, while $r_{\mathrm{eff}}$ follows an adiabatic profile. The retrieved values of $N$ using the homogeneous or the inhomogeneous model deviate within several percent. Further studies show that the inhomogeneous model represents nature more realistically compared to the homogeneous assumption (Boers et al., 1998; Brenguier et al., 2000). These methods often use the dependence of $\tau$ on $N$ to connect cloud microphysical and radiative properties. However, so far no operational satellite

products of $N$ are available and can have uncertainties of up to $80\,\%$ (Grosvenor et al., 2018b).

Instead of using $\tau$ in the remote sensing of $N$, $LWP$ from passive microwave sensors can be exploited (Minnis et al., 2011). This has the advantage that $LWP$ is determined at wavelengths, which are not influence by aerosol, sun-glint, or three-dimensional (3D) radiative effects. Further on, active remote sensing techniques have been applied to derive $N$, e.g., by Austin and Stephens (2001), Mace et al. (2016), who combined $r_{\mathrm{eff}}$ vertical profiles derived by cloud radar observations and $\tau$

obtained from passive solar remote sensing. Nevertheless, a disadvantage of the radar is that the radar reflectivity $Z$ is mainly determined by large cloud droplets, which biases the results.

The dependence of $\tau$ on $N$ is investigated by Quaas et al. (2009) using satellite measurements. The correlations of $\tau$ and $N$, obtained by satellite are weaker compared to aircraft remote sensing results or in-situ measurements, which primarily is due to the large-scale averaging of the satellite measurement (McComiskey and Feingold, 2008). Analyzing satellite measurements

of large-scale averaged $N$ and $\tau$ in different thermodynamic conditions and, therefore, varying $LWP$, updraft velocity and





aerosol particle concentrations, mask the effect of $N$ on $\tau$. As a result, parameterizations derived from satellite observations are note well suited for trade-wind cumulus with their natural variability.

Airborne remote sensing may bridge the gap between in-situ and satellite measurements, as it allows to sample individual clouds under specific conditions and to cover a sufficiently large area to quantify the natural variability of $N$, $r_{\text{eff}}$, and $LWC$.

Here a method is proposed to combine passive and active airborne remote sensing measurements of cloud vertical profiles of microphysical parameters and cloud radiative properties. Measurements of upward irradiance $I_\lambda^\uparrow$ by the Spectral Modular Airborne Radiation measurement sysTem (SMART) are used to determine $\tau$, $r_{\text{eff}}$, and the thermodynamic phase of cloud water close to the cloud top. Observations of the High Altitude and LOng range research aircraft Microwave Package (HAMP), which comprises of a multi-channel microwave radiometer and a cloud radar. HAMP provides $LWP$ and cloud geometry,

which allows to discriminate between precipitating and non-precipitating clouds. Furthermore, an alternative retrieval to determine $r_{\text{eff}}$ from the spectromter-microwave combination of SMART and HAMP is developed and tested. Lidar measurements of the Water Vapour Lidar Experiment in Space (WALES) are additionally implemented to determine the cloud top height $h_{\text{top}}$, while dropsondes provide estimates of the cloud base height $h_{\text{base}}$.

This paper is structured as follows. In Section 2 the sensitivity of the cloud top reflectivity $\mathcal{R}$ and cloud top albedo $\alpha$ of typical

trade-wind cumuli with respect to changes of $N$ is quantified. This is investigated to access the required accuracy of $N$ retrievals and the cloud regime most sensitive to $N$. The remote sensing instruments utilized in this study are introduced briefly in Section 3. In Section 4 the retrieval of the optical properties and the cloud filtering is described. Subsequently, the three different methods to determine $N$ are presented in Section 5 and applied to synthetic measurements and two exemplary cases of trade wind cumulus. Resulting values of $N$ are correlated with measured cloud top reflectivity $\mathcal{R}$, separated for different

thermodynamic conditions (binned $LWP$), to show the possibility to obtain parameterizations for the Twomey effect.





## 2 Sensitivity of the Twomey Effect for Different Cloud Regimes

Cloud optical and cloud microphysical properties depend on the composition and number of aerosol particles as described by the Twomey effect (Twomey, 1977). A sensitivity study was performed to quantify the dependence of cloud top albedo $\alpha$ or cloud top reflectivity $\mathcal{R}$ on $N$ and $LWP$. To determine which cloud regimes, defined by combinations of $N$ and $LWP$, is

most sensitive to changes of N.

To quantify the Twomey effect for trade wind cumulus with different $LWP$, radiative transfer simulations (RTS) with the radiative transfer package libRadtran 2.0.2 (Emde et al., 2016) are performed. The solar cloud top albedo was calculated for a homogeneous liquid water cloud located between 1000 m and 1500 m and a solar zenith angle $\vartheta$ of $5°$. Liquid water path is varied in a range between $10\,\mathrm{g\,m^{-2}}$ and $200\,\mathrm{g\,m^{-2}}$ which was based on in-situ measurements of shallow trade wind cumulus

by Siebert et al. (2013).

Figure 1a shows simulated $\alpha$ for all combinations of $N$ and $LWP$. For constant $LWP$ and increasing $N$ (decreasing $r_{\mathrm{eff}}$), $\alpha$ increases which is described by the Twomey effect. However, this sensitivity is not equal for the different $LWP$. For constant $N$ and increasing $LWP$ (increasing $r_{\mathrm{eff}}$), $\alpha$ increases with different rates for $N$. This clearly shows that different cloud regimes react differently to the Twomey effect following Eq. (1), and therefore, $LWP$, $N$, and $r_{\mathrm{eff}}$ have to be considered to

parameterize the radiative properties of trade-wind cumuli.

To quantify the Twomey effect for different cloud regimes, the sensitivity $\zeta$ is calculated:

$$\zeta(LWP, r_{\mathrm{eff}}, N) = \frac{\mathrm{d}\alpha(LWP, r_{\mathrm{eff}}, N)}{\mathrm{d}N}, \tag{2}$$

which represents the change of $\alpha$ with respect to an increase of $N$ and is given in units of $\mathrm{cm^3}$.

Figure 1b displays $\zeta$ as a function of $N$ for different $LWP$. In general, $\zeta$ decreases with increasing $N$. Clouds with low $LWP$

and low $N$ have a higher $\zeta$ compared to clouds with higher $LWP$ but same $N$. The highest $\zeta$ is obtained for clouds with the lowest $LWP$ of $10\,\mathrm{g\,m^{-2}}$, while thicker clouds with the highest $LWP$ of $200\,\mathrm{g\,m^{-2}}$ have the lowest Twomey effect, lowest $\zeta$.

In Fig. 1c the sensitivity $\zeta$ is shown as a function of $r_{\mathrm{eff}}$ for clouds of different $LWP$. Assuming a constant $LWP$, the effective radius determines $N$ or vice versa following

$$r_{\mathrm{eff}} = \sqrt[3]{\frac{3}{4} \cdot \frac{LWP}{\rho_{\mathrm{w}} \cdot \pi \cdot \mathrm{d}z \cdot N} \cdot k^{-3}}. \tag{3}$$

For all $LWP$ cases the sensitivity increases with increasing $r_{\mathrm{eff}}$ (decreasing $N$). This agrees with Fig. 1b where low $N$ have the highest $\zeta$. Clouds with lower $LWP$ show higher $\zeta$ and, therefore are more sensitive to changes of $r_{\mathrm{eff}}$ compared to clouds with higher $LWP$.

In Fig. 1d $\zeta$ is plotted as a function of $\tau$, which is calculated using Eq. (1) from $LWP$, $N$, and $r_{\mathrm{eff}}$ used in the simulations. For

all clouds with different values of $LWP$, $\zeta$ decreases with increasing $\tau$. This implies that changes in $N$ have larger effects on $\alpha$ for clouds with low $\tau$. As a result, optically thin clouds with low $N$ and large $r_{\mathrm{eff}}$, which is the typical character of shallow trade wind cumulus, are subject to the strongest Twomey effect. Therefore, the Twomey effect on trade wind cumulus is highly

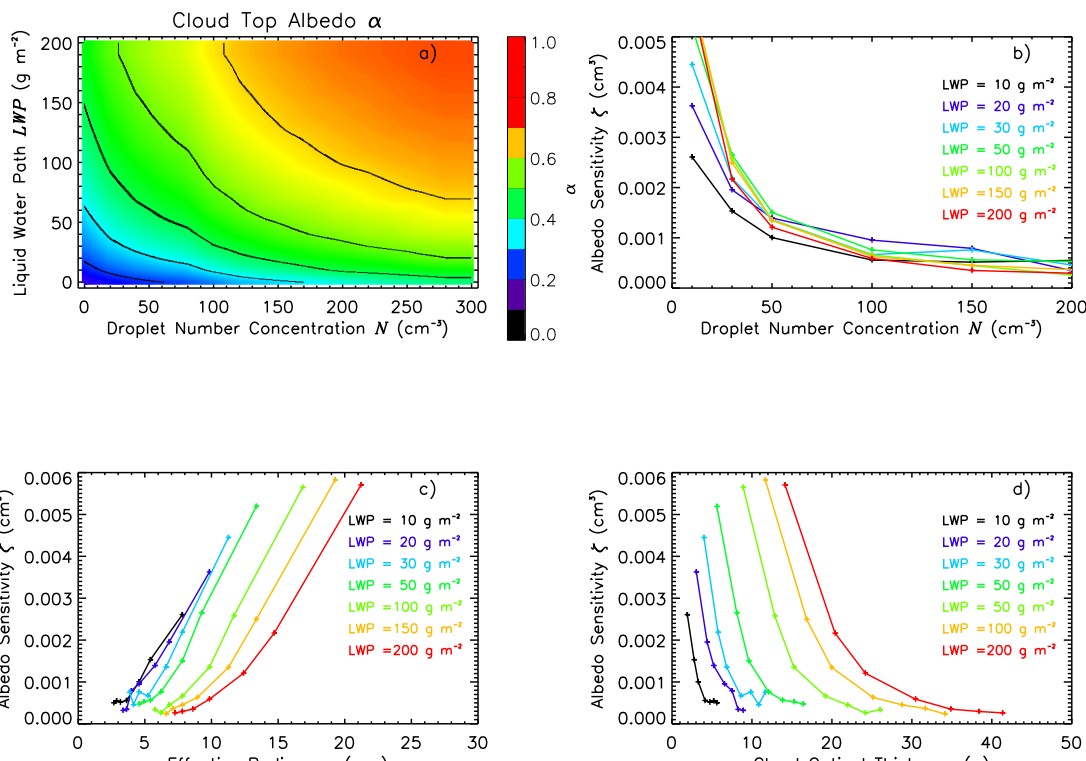

**Figure 1.** Simulations for a liquid water cloud between 1000 m and 1500 m with liquid water path $LWP$ from $10\,\mathrm{g\,m^{-2}}$ to $200\,\mathrm{g\,m^{-2}}$ and for a solar zenith angle $\vartheta$ of $5°$. The simulations are integrated over a wavelength range from 250 nm to 2500 nm. Panel a) shows cloud top albedo $\alpha$ for combinations of the cloud droplet number concentration $N$ and $LWP$. Panel b) shows cloud top albedo sensitivity $\zeta$ as a function of $N$ for different $LWP$. Panel c) and d) display $\zeta$ as a function of effective radius $r_{\mathrm{eff}}$ and cloud optical thickness $\tau$, respectively.





relevant for NWP and GCMs.

The simulations further illustrate the challenge of estimating $\alpha$ of shallow trade wind cumulus by satellite remote sensing. Typically, satellite retrievals of $N$ can have uncertainties in the range of up to $80\,\%$ (Grosvenor et al., 2018b). For clouds with low $N$, e.g., $30\,\mathrm{cm}^{-3}$, and $LWC = 0.1\,\mathrm{g\,m}^{-3}$ the concentration of $N$ might be biased by up to $\pm 23\,\mathrm{cm}^{-3}$. This would result in

a bias of $\alpha$ of $\pm 0.08$ ($80\,\mathrm{W\,m}^{-2}$ increased cloud forcing for $1000\,\mathrm{W\,m}^{-2}$ insolation). For clouds with higher $N$ of $200\,\mathrm{cm}^{-3}$ the retrieval uncertainties of $N$ naturally increase in absolute values ($\Delta N = \pm 156\,\mathrm{cm}^{-3}$) and still lead to an almost similar uncertainty of $\alpha = \pm 0.07$ even though $\zeta$ is reduced for clouds with higher $N$. This shows, that retrievals of $N$ need to be improved, in order to improve global estimates of $N$ and to reduce the uncertainty of $\alpha$ calculations in NWP and GCM.

## 3   Observations and Instrumentation

Convective shallow trade wind cumulus have been observed by airborne remote sensing during the second Next Generation Remote Sensing for Validation Studies (NARVAL-II) campaign between 8 and 31 August 2016 (Stevens et al., 2018). The High Altitude and Long Range Research Aircraft (HALO) operated from Barbados mostly flying eastward into an area dominated by shallow trade wind cumulus mostly unaffected by anthropogenic influences. HALO was equipped with a set of passive and active remote sensing instruments. Reflected solar radiation was measured by the passive instruments SMART (Wendisch et al.,

2001, 2016) and Munich Aerosol Cloud Scanner (specMACS) (Ewald et al., 2016), while radiation emitted in the microwave spectral range was measured by the HALO Microwave Package (HAMP). For active remote sensing, HAMP included an cloud radar (Mech et al., 2014). Lidar observations by the WAter vapor Lidar Experiment in Space (WALES) completed the cloud remote sensing instrumentation. WALES measures the backscatter coefficient and depolarization at 532 nm and 1064 nm wavelength, and contains a high spectral resolution lidar channel at 532 nm wavelength (Wirth et al., 2009). Additionally,

dropsondes were released from HALO.

All instruments were orientated into nadir direction and synchronized in time. However, the different Field-of-Views (FOV) of the instruments cause a systematic difference in the observed time series. All measured and retrieved quantities from SMART, HAMP, WALES, and the dropsondes are listed in Table 1.

### 3.1   Spectral Modular Airborne Radiation measurement sysTem

During NARVAL-II, SMART measured the spectral upward $F_\lambda^\uparrow$ and downward irradiance $F_\lambda^\downarrow$, as well as spectral upward radiance $I_\lambda^\uparrow$. Each quantity was recorded with two separate Zeiss grating spectrometers, one for the visible (VIS) range from 300 nm to 1000 nm and a second one for sampling the near-infrared (NIR) range from 900 nm to 2200 nm. By merging the spectra, about $97\,\%$ of the solar spectrum is covered (Bierwirth et al., 2009). The spectral resolution defined by the full width at half maximum is 2 - 3 nm for the VIS spectrometer and 8 - 10 nm for the NIR spectrometer.

The radiance optical inlet of SMART has an opening angle of $2°$. The sampling time $t_{\mathrm{int}}$ was set to 0.5 s. For an average aircraft ground-speed of about $220\,\mathrm{m\,s}^{-1}$ and a distance of 10 km between cloud top and the aircraft this results in a FOV of about 100 m x 120 m for an individual pixel.



**Table 1.** Measured and retrieved quantities from SMART, HAMP, WALES, and the dropsondes.

| Instrument | Measured / retrieved quantity | Variable | Unit |
|---|---|---|---|
| SMART | Upward radiance | $I_\lambda^\uparrow$ | $\mathrm{W\,m^{-2}\,sr^{-2}}$ |
| | Cloud optical thickness | $\tau$ | - |
| | Effective radius | $r_{\mathrm{eff}}$ | $\mu\mathrm{m}$ |
| | Liquid water path | $LWP_\mathrm{A}$ | $\mathrm{g\,m^{-2}}$ |
| HAMP | Liquid water path | $LWP_\mathrm{B}$ | $\mathrm{g\,m^{-2}}$ |
| | Radar reflectivity | $Z$ | dBz |
| WALES | Cloud top height | $h_{\mathrm{top}}$ | m |
| Dropsondes | Temperature | $T$ | °C |
| | Dew-point temperature | $T_\mathrm{d}$ | °C |
| | Lifting condensation level | $h_{\mathrm{LCL}}$ | m |

The optical inlets for $F_\lambda^\uparrow$ and $F_\lambda^\downarrow$ mainly consist of integrating spheres, which collect direct and scatter solar radiation from the upper or lower hemisphere. During NARVAL-II the upward-looking inlet was equipped with an active stabilization platform to ensure horizontal alignment of the sensor, which is crucial as $F_\lambda^\downarrow$ refers to a horizontal plane (Wendisch et al., 2001).

Prior and after NARVAL-II, SMART was radiometrically calibrated in the laboratory using certified calibration standards

traceable to the National Institute of Standards and Technology (NIST). A secondary calibration by a mobile standard was applied during the campaigns to track potential changes of the instrument sensitivity. The total measurement uncertainty of downward irradiance $F_\lambda^\downarrow$ and upward radiance $I_\lambda^\uparrow$ for typical conditions and observations of of shallow cumulus is about $5.4\,\%$ for the VIS and $8.4\,\%$ for the NIR range, which is composed of individual errors due to the spectral calibration, the spectrometer noise and dark current, the primary radiometric calibration (Brückner et al., 2014).

## 3.2   HALO Microwave Package

HAMP is a combination of a passive microwave radiometer and an active cloud radar specifically designed for the operation on HALO (Mech et al., 2014). The microwave radiometer includes 26 frequency channels between $22.24\,\mathrm{GHz}$ and $183.31\,\mathrm{GHz} \pm 12.5\,\mathrm{GHz}$. The brightness temperature (BT) measured along the $22.24\,\mathrm{GHz}$ and $183.31\,\mathrm{GHz}$ rotational water vapor lines provide accurate information on the total column water vapor (Schnitt et al., 2017) and information on its vertical

distribution. Liquid water emission increases roughly with the frequency squared. By combining BT in window channels, i.e. $31.4\,\mathrm{GHz}$ and $90\,\mathrm{GHz}$, mostly affected by liquid water with those sensitive to water vapor the $LWP$ can be retrieved. This principle is also employed by satellite instruments which provide global climatologies of $LWP$, which suffer from the coarse footprint of a few 10ths of kilometer (Elsaesser et al., 2017).

The statistical $LWP$ retrieval is based on a large variety of atmospheric profiles with differently structured warm clouds as

training data composed from the dropsondes as described by Schnitt et al. (2017). Synthetic BT are simulated from these pro-





files and subsequently used to fit a multi-parameter linear regression model employing higher order terms following Mech et al. (2007). Testing the retrieval algorithm on an independent subsample provides an accuracy of about $30\,\mathrm{g\,m^{-2}}$ for $LWP$ values below $500\,\mathrm{g\,m^{-2}}$.

The cloud radar MIRA-36 operates at a frequency of 36 GHz and has a similar horizontal resolution as the $LWP$ of about 1000 m and a temporal resolution of 1 s. Vertical profiles are divided into 30 m bins (Mech et al., 2014). The radar provides different parameters linked to the cloud microphysical properties including the radar reflectivity $Z$, the linear depolarization, and the Doppler velocity and the spectral width of the droplet size distribution. Note, that the latter two are affected by the relative motion of the aircraft to the wind and the antenna width (Mech et al., 2014) and are not used here.

Radar reflectivity is the sixth moment of the drop size distribution and therefore is strongly influenced by the largest particles. In order to calculate the $LWC$, i.e. proportional to the third moment of the DSD, from $Z$ so-called Z-LWC relations are used which are typically derived from in-situ measurements. According to Khain et al. (2008) there is quite some variability involved and as soon as the transition to drizzle sets in the relation can be off by orders of magnitude. Here the Z-LWC relation

$$LWC_{\mathrm{p}} = LWP \cdot \frac{\sqrt{Z_p}}{\sum_{j=1}^{j=M} \sqrt{Z_j} \cdot \Delta h} \tag{4}$$

following (Frisch et al., 2000) is used to derive vertical profiles of $LWC$. With the binned $LWC_{\mathrm{p}}$ at height gate p resulting from the vertical resolution of the radar, the $LWP$ of the cloud, is distributed by the weighting of $Z_p$ ($Z$ at height gate p) and $\sum_{j=1}^{j=M} \sqrt{Z_j} \Delta h$ the sum of the $Z$, over all height gates where a cloud was present.

Brightness temperatures and radar reflectivity profiles are described in more detail by Konow (2018).

## 3.3 Water Vapour Lidar Experiment in Space (WALES)

The DIfferential Absorption Lidar (DIAL) called WALES operates at four wavelengths near 935 nm to measure atmospheric water vapor. It provides water-vapor mixing ratio profiles covering the whole atmosphere below the aircraft. The system also contains additional aerosol channels at 532 nm and 1064 nm with depolarization. At 532 nm WALES uses the high-spectral resolution technique, which distinguishes molecular from particle backscatter, to make direct extinction measurements. Within this study only the aerosol channels are used to provide information on the cloud top height. The ranging resolution of the instrument is 15 m. Together with the flight altitude inferred from the HALO on-board positioning system and an appropriate attitude correction the accuracy of cloud top height detection is about 20 m relative to the geoid. Over the sea this can be verified by looking at the surface reflex.

The laser has a beam divergence of 1 mrad which leads to an illuminated spot of 10 m diameter on ground at a flight altitude of 10 km. Laser pulses are emitted with a repetition rate of 100 Hz, but 20 signals are averaged on board to improve the signal to noise ratio, resulting in an along flight track resolution of 44 m at $200\,\mathrm{m\,s^{-1}}$ aircraft speed. Thus the horizontal resolution is considerably smaller than that of SMART and HAMP. Along track this can be easily taken into account by further signal averaging.





## 4 Measurement Analysis

Trade wind cumulus mostly appear randomly distributed with a tendency to form self-organizing structures (Bony et al., 2015). Typically, the vertical cloud extend is larger than the horizontal extent within an individual cell. This is in strong contrast to stratiform cloud fields to which common retrieval techniques of $N$ are usually applied. Clouds smaller than pixel size covered

by the FOV bias the retrieval of the microphysical properties significantly (Oreopoulos and Davies, 1998a, b). With respect to the total cloud volume, trade wind cumuli are stronger effected by turbulent mixing compared to stratiform clouds. The dominance of small-scale cumulus during NARVAL-II, mostly ranging in the horizontal size of a few hundred meters, results in very heterogeneous cloud scenes. This induces several challenges with respect to cloud masking and RTS. Several filters introduced below are applied to the data in order to minimize heterogeneity effects.

### 4.1 Cloud Mask and Precipitation Flag

#### 4.1.1 Cloud Mask

To distinct between cloud and cloud-free measurements over ocean surfaces is linked with the difference in the spectral reflectivity is analyzed. The ratio $\chi$ of $I_\lambda^\uparrow$ between 858 nm and 648 nm wavelength is calculated in analogy to the MODIS cloud

mask (Platnick et al., 2013) by:

$$\chi = \frac{I_{858}}{I_{648}}. \tag{5}$$

The cloud mask is based on the relative intensity of $I_\lambda^\uparrow$ and $\chi$. Therefore, a single measurement can be identified as cloudy when only a part of the SMART FOV with 100 m x 120 m is covered by a cloud. Masking each measurement point as cloudy or cloud-free, the cloud length $l_{cld}$ is determined. By counting the number $n$ of consecutive cloud masked measurements.

Multiplying with the flight speed $v_{aircraft}$ and the constant integration time of SMART of $t_{int} = 0.5$ s, the cloud length is calculated with:

$$l_{cld} = n \cdot t_{int} \cdot v_{aircraft}. \tag{6}$$

For $v_{aircraft} \approx 220 \, \mathrm{m\,s^{-1}}$ the smallest cloud size which can be resolved is in the range of 120 m along flight track.

The length of trade wind cumulus can be shorter than the FOV of SMART. To identify such cases, an additional homogeneity

cloud flag (HCF) is introduced. The cloud is considered as homogeneous (HCF is true) when a single observation and two previous and two following measurements are masked as cloud too. For clouds not surrounded by at least two cloudy pixel, the HCF is set to false. Therefore, the HCF identifies clouds that are large enough to fill the FOV of SMART, HAMP, and WALES at the same time.



### 4.1.2 Precipitation Flag

Precipitation is identified using the radar reflectivity $Z$. Measurements are considered to be affected by precipitation when $Z$ exceeds a threshold of $Z <$ -20 dBz within 50 m to 200 m above sea level (Schnitt et al., 2017). This allows to discriminate precipitation events, which affect the $LWP$ measured by the microwave radiometer and retrieved by SMART. Using radar observations to identify rain is prone to deficiencies. The radar reflectivity is very sensitive to the droplet diameter $D$ by $Z \propto D^6$ and only droplets with sufficient size are detected. Hence, measurements with low $Z$ below the detection threshold can still contain slight drizzle.

### 4.2 Retrieval of Cloud Optical Thickness and Effective Radius

Based on the reflected solar radiance $I_\lambda^\uparrow$ measured by SMART, a retrieval of $\tau$ and $r_{\mathrm{eff}}$ is performed, applying the radiance ratio method proposed by Werner et al. (2013). The use of radiance ratios at two different wavelength partly reduces the uncertainties by the radiometric calibration of SMART. For the wavelength ratio used here, an uncertainty of $6\,\%$ is assumed. Additionally, the use of ratios increases the retrieval sensitivity with respect to $r_{\mathrm{eff}}$ by clearly separating the dependency of $I_\lambda^\uparrow$ on $\tau$ and $r_{\mathrm{eff}}$ and, therefore, the retrieval accuracy. Forward simulations of reflected spectral radiance $I_\lambda^\uparrow$ were carried out with the libRadtran 2.0.2 package (Emde et al., 2016). The Fortran 77 discrete ordinate radiative transfer solver version 2.0 (FDIS-ORT 2) after Stamnes et al. (2000) is used. The extraterrestrial $F_\lambda^\downarrow$ is given by Gueymard (2004) and a marine aerosol profile after Shettle (1989) is selected. Vertical profiles of temperature, pressure, and humidity are obtained from radiosonds released at the Bridgetown International Airport. For the optical properties of liquid water droplets, Mie calculations are performed.

The optical thickness and $r_{\mathrm{eff}}$ are determined by a modified Look-Up-Table (LUT) method after Nakajima and King (1990). The optical thickness is determined at 870 nm. The effective radius is derived with the radiance ratio method, using a ratio of measurements at 1645 nm and 1050 nm and denoted with subscript "A".

Clouds which do not cover the entire FOV of SMART bias the retrieved optical properties, because they are in direct contradiction to the assumption of plane parallel clouds used in the RTS (Oreopoulos and Davies, 1998a, b). Lower values of $I_\lambda^\uparrow$ bias $\tau$ towards lower values, whereas $r_{\mathrm{eff}}$ is shifting to larger droplet sizes (Cahalan et al., 1995). Further on, the heterogeneous structure of trade wind cumulus may cause 3D radiative effects, like shadowing of parts of the cloud by nearby cloud-towers or enhanced reflectivity due to additional reflection into the FOV. These effects may also bias the retrieval of $\tau$ and $r_{\mathrm{eff}}$ and the calculation of $N$. Therefore, the HCF filter is applied to exclude measurements that are influenced by these processes. However, due to the low vertical extend of shallow trade wind cumulus these 3D radiative effects are assumed to be not pronounced for the cases analyzed here.

Finally, $LWP$ is obtained directly from the parametrization within libRadtan which calculates "LWP" on the basis of $\tau$ and $r_{\mathrm{eff}}$ similar to Eq. 1. Liquid water path derived from SMART is denoted with subscript "A". In case of cloud heterogeneity, sun-glint or 3D radiative effects, the retrieval of $\tau$ is very likely biased. Following Eq. 1, a bias of $\tau$ also influences the retrieval of $r_{\mathrm{eff}}$ and therefore, $LWP$. To omit these effects, measurements of the $LWP$ from HAMP (denoted with subscript "B") are



applied in the forward model of the cloud retrieval. Liquid water path from microwave radiometers are obtained from wavelength not influenced by sun-glint or 3D radiative effects. Using $LWP$ from HAMP as a precondition, the LUTs reduce to one absorbing wavelength sensitive to $r_{\mathrm{eff}}$. Therefore, the non-linear dependence between $\tau$ and $r_{\mathrm{eff}}$ is removed and the retrieval becomes more reliable. Retrieved $r_{\mathrm{eff}}$ from combined passive solar radiance and microwave measurements are denoted with

subscript "B".

## 5   Retrieval of Cloud Droplet Number Concentration

The retrieval of $N$ from remote sensing observations is generally based on the relation proposed by Brenguier et al. (2000) and Wood (2006), which links $N$ of a stratiform cloud to $\tau$ and $r_{\mathrm{eff}}$ by:

$$N_{\mathrm{A}} = \frac{\sqrt{10}}{4 \cdot \pi \cdot \sqrt{\rho_{\mathrm{w}}}} \cdot \sqrt{f_{\mathrm{ad}} \cdot \Gamma_{\mathrm{ad}}} \cdot \frac{\sqrt{\tau}}{\sqrt{r_{\mathrm{eff,A}}^{5}}}. \tag{7}$$

The technique assumes an adiabatic vertical cloud profile, where temperature linearly decreases and $LWC$ linearly increases with height. An adiabatic profile implies that the total water mass mixing ratio of the cloud is conserved. This is true when: (i) no water is removed from the cloud (no precipitation or fallout), (ii) no entrainment of dryer air at the cloud edges occurs, and (iii) no evaporation from precipitation happens. As a result, the proposed method should be applied to non-precipitating clouds only, which do not undergo strong vertical convection and mixing. A vertically constant $N$ throughout the cloud layer

is assumed. This assumption is verified for stratiform clouds and shallow trade wind cumulus by in-situ measurements, e.g., Reid et al. (1999) and Wendisch and Keil (1999). The main reason for the vertically constant $N$ is the determination by the amount of available CCN at cloud base and their potential to form cloud droplets depending on the degree of supersaturation, which is controlled by temperature, entrainment of dry air and updraft velocity.

The $k$-parameter,

$$k = \left( \frac{r_{\mathrm{vol}}}{r_{\mathrm{eff}}} \right)^{3} \tag{8}$$

relating the effective radius $r_{\mathrm{eff}}$ and the volumetric radius $r_{\mathrm{vol}}$, is set to $k = 0.8$ for marine clouds following the suggestion by Martin et al. (1994) and Pontikis (1996). Depending on the cloud type the $k$-parameter can vary by $\pm 0.1$ (Martin et al., 1994). With help of cloud properties retrieved by airborne remote sensing Eq. (7) can be applied in different complexity to derive $N$. Three methods are proposed in the following. Method A uses only SMART data, while method B additionally includes HAMP

observations of $LWP$, whereas method C also involves measurements by WALES. The obtained parameters and assumptions used by the different methods are summarized in Table 2.

### 5.1   Method A: Based on Cloud Optical Thickness and Effective Radius

Method A follows the traditional satellite approach to feed Eq. (7) with $\tau$ and $r_{\mathrm{eff}}$ obtained by a single passive remote sensing

instrument. Here the $\tau_{\mathrm{A}}$ and $r_{\mathrm{A}}$ retrieved by SMART are applied. The degree of adiabacity is assumed to be 1. This implies,



**Table 2.** Overview of the cloud droplet number concentration retrievals and applied measurements, retrieval parameters and assumptions.

| Method | | A | B | C |
|---|---|---|---|---|
| **Instruments and Parameters** | | | | |
| | SMART | $\tau, r_{\mathrm{reff,A}}$ | $r_{\mathrm{reff,B}}$ | $r_{\mathrm{reff,B}}$ |
| | HAMP | $\times$ | $LWP$ | $LWP$ |
| | WALES | $\times$ | $\times$ | $f_{\mathrm{calc}}$ |
| **Assumptions** | | | | |
| | adiabatic cloud-profile | $\checkmark$ | $\checkmark$ | $\times$ |
| | adiabatic change of LWC | $f_{\mathrm{ad}} \cdot \Gamma_{\mathrm{ad}} = 2.5 \cdot 10^{-3}\,\mathrm{g\,m^{-3}\,m^{-1}}$ | | $\Gamma_{\mathrm{calc}}$ |
| | $k$-parameter | $k = 0.8$ | $k = 0.8$ | $k = 0.8$ |
| | const. $N$ | $\checkmark$ | $\checkmark$ | $\checkmark$ |
| | deep convection | $\times$ | $\times$ | $\times$ |
| | cloud homogeniety | $\checkmark$ | $\checkmark$ | $\checkmark$ |
| | precipitation | $\times$ | $\times$ | $\times$ |
| | min. hori. size | $\approx 150\,\mathrm{m}$ | $\approx 150\,\mathrm{m}$ | $\approx 150\,\mathrm{m}$ |

that for trade wind cumuli, which are typically sub-adiabatic, the estimated $N$ is potentially biased. However, similar retrieval are frequently applied to observations from satellite such as MODIS (Grosvenor et al., 2018b).

## 5.2 Method B: Based on Liquid Water Path and Effective Radius

For adiabatic clouds, Eq. (1) can be solved analytically, which results in a relation that directly links $LWP$ to $\tau$ and $r_{\mathrm{eff}}$:

$$LWP = \frac{5}{9} \cdot \rho_{\mathrm{w}} \cdot \tau \cdot r_{\mathrm{eff}} \tag{9}$$

following (Brenguier et al., 2000). Equation (9) allows to apply Eq. (7) with an independent measure of $LWP$ instead of $\tau$ to calculate $N$. As given by Wood (2006) combining Eq. (7) and Eq. (9) leads to:

$$N_{\mathrm{B}} = \frac{3 \cdot \sqrt{2}}{4 \cdot \pi \cdot \rho_{\mathrm{w}}} \cdot \sqrt{f_{\mathrm{ad}} \cdot \Gamma_{\mathrm{ad}}} \cdot \frac{\sqrt{LWP_{\mathrm{B}}}}{r_{\mathrm{eff,B}}^3}. \tag{10}$$

In method B, $LWP$ measurements by HAMP and derived $r_{\mathrm{eff,B}}$ from the combined SMART microwave-radiometer retrieval are applied. The results are denoted with $N_{\mathrm{B}}$. Exchanging $r_{\mathrm{eff,A}}$ by $r_{\mathrm{eff,B}}$ takes into account that $LWP$ is determined from HAMP only. This makes the retrieval independent on $\tau$ derived by SMART and therefore less sensitive to effects by sun glint.





### 5.3 Method C: Based on Liquid Water Path, Effective Radius, and Cloud Geometric Thickness

Equations (7) and (10) assume constant values of $f_{\mathrm{ad}}$ and $\Gamma_{\mathrm{ad}}$. Therefore, in method A and B the adiabatic profile of $LWC$ follows the maximum, theoretically possible profile under which liquid water is released due to condensation from upward motion in the atmosphere.

In-situ measurements of stratocumulus and shallow cumulus clouds, like trade wind cumulus, indicate that a majority of cloud profiles do not follow this adiabatic assumption (Wendisch and Keil, 1999; Merk et al., 2016). In most cases the profiles are sub-adiabatic, meaning a reduced increase of $LWC$ with height, mostly due to entrainment and mixing from dry air at the cloud edges. Entrainment and mixing reduce $f_{\mathrm{ad}}$ but not necessarily $N$. When convection and mixing is moderate, an equilibrium between the droplets and the surrounding air can be assumed. Therefore, the reduced (super-)saturation due to entrainment at

the cloud edges will cause a shrinking of the droplets but not their complete vanishing. To account for a sub-adiabatic increase of $LWC$ with height in method C, $f_{\mathrm{ad}} \cdot \Gamma_{\mathrm{ad}}$ is replaced by observations. The increase of $LWC$ with height $\Gamma_{\mathrm{calc}}$ is calculated by:

$$\Gamma_{\mathrm{calc}} = \frac{2 \cdot LWP_{\mathrm{B}}}{(\mathrm{d}z)^2} \tag{11}$$

with $LWP_{\mathrm{B}}$ obtained by the microwave radiometer. The cloud geometric thickness $\mathrm{d}z = h_{\mathrm{top}} - h_{\mathrm{LCL}}$ is estimated from a

combination of the WALES $h_{\mathrm{top}}$ observations and $h_{\mathrm{LCL}}$ from dropsondes.

WALES can only derive cloud top height $h_{\mathrm{top}}$ when the laser is attenuated by clouds with high $\tau$. As a result, the lidar signal is attenuated soon and the cloud base height is not detectable. Therefore, $h_{\mathrm{base}}$ is determined separately from dropsondes, which represent the large-scale thermodynamic structure of the atmosphere. Using the temperature $T$ and dew point temperature $T_{\mathrm{d}}$ at the two lower most points, the lifting condensation level with $h_{\mathrm{LCL}} \approx 125 \cdot (T - T_{\mathrm{d}})$ is approximated (Espy, 1836). Nevertheless,

uncertainties of estimated $h_{\mathrm{LCL}}$ from dropsondes are in the range of $\pm 35$ m not considering additional uncertainties caused by the assumptions in the equation (Romps, 2017). Using the estimated $\Gamma_{\mathrm{calc}}$, Eq. (10) changes to:

$$N_{\mathrm{C}} = \frac{3 \cdot \sqrt{2}}{4 \cdot \pi \cdot \rho_{\mathrm{w}}} \cdot \frac{LWP_{\mathrm{B}}}{\mathrm{d}z \cdot r_{\mathrm{eff,B}}^3}. \tag{12}$$

Calculations of $N$ using the estimated $\mathrm{d}z$ and resulting $\Gamma_{\mathrm{calc}}$ are denoted with $N_{\mathrm{C}}$. Alternatively, the cloud top height and cloud base height $h_{\mathrm{base}}$ can be determined with the radar for all cloud cases where: (i) the cloud droplets are large enough to

produce a detectable radar echo, and (ii) no precipitation is present.

### 5.4 Simulated Synthetic Measurements

To systematically test the potential of the proposed synergistic retrieval methods, synthetic measurements of spectral upward radiance $I_{\lambda,\mathrm{syn}}^{\uparrow}$ are created. In that way, the three different methods are compared omitting the influence by measurement errors.

Further on, varying environmental conditions, like sea surface albedo, heterogeneous cloud conditions, and 3D cloud radiative effects do not influence the systematic comparison of the retrieval methods. The comparison is based on retrieved cloud droplet





number concentration $N$ with methods A, B, and C and $N_{\mathrm{cld}}$ calculated from the model clouds serving as truth value.

Six synthetic clouds are simulated. Their respective parameters are listed in Table 3. Cloud droplet number concentrations $N_{\mathrm{cld}}$ of $50\,\mathrm{cm}^{-3}$, $100\,\mathrm{cm}^{-3}$, and $200\,\mathrm{cm}^{-3}$ represent the typical range of pristine shallow trade wind cumulus (Siebert et al., 2013). For each $N_{\mathrm{cld}}$ an adiabatic and a sub-adiabatic cloud profile was set up. Cloud base height is 500 m and cloud top

height is 1000 m. For all cloud cases a linear increase of $LWC$ and a constant $N_{\mathrm{cld}}$ with height are assumed. In the adiabatic cases (I, III, V) a $LWP$ of $362\,\mathrm{g\,m}^{-2}$ and an adiabatic increase of $LWC$ with height $\Gamma_{\mathrm{ad}}$ of $2.9\cdot10^{-6}\,\mathrm{kg\,m}^{-3}\,\mathrm{m}^{-1}$, for a surface temperature of $\approx 30^{\circ}\,\mathrm{C}$ are used. For the sub-adiabatic cases (II, IV, VI) $\Gamma$ is set to $\Gamma_{\mathrm{ad}}\cdot0.6 = 1.7\cdot10^{-6}\,\mathrm{kg\,m}^{-3}\,\mathrm{m}^{-1}$ representing a cloud which follows $\Gamma_{\mathrm{ad}}$ by $60\%$ and leads to a $LWP$ of $217\,\mathrm{g\,m}^{-2}$. To calculate the volumetric radius $r_{\mathrm{vol}}(z)$, the cloud profiles are divided into 20 layers of equal thickness of 25 m. For each layer the parameterization of Martin et al.

(1994) is applied:

$$r_{\mathrm{vol}}(z) = \sqrt[3]{\frac{3\cdot LWC(z)}{4\cdot\rho_{\mathrm{w}}\cdot\pi\cdot N_{\mathrm{cld}}}}. \tag{13}$$

In the radiative transfer model, the effective radius $r_{\mathrm{eff}}$ is used to determine the optical properties of the cloud particles instead of the volumetric radius $r_{\mathrm{vol}}$. To convert $r_{\mathrm{vol}}(z)$ into $r_{\mathrm{eff}}(z)$ a $k$ of 1.0 is applied, what considers the monodisperse droplet size distribution used in the model clouds. The synthetic measurements of $I^{\uparrow}_{\lambda,\mathrm{syn}}$ are calculated with the same simulation set-up as

for the cloud retrieval described in Section 4.2.

Simulated synthetic measurements of $I^{\uparrow}_{\lambda,\mathrm{syn}}$ are applied to the retrieval method of $\tau$, $r_{\mathrm{eff}}$, and $N$ of Section 5. All three methods A, B, and C are applied and results are denoted with additional subscript "R". The true values of $\tau$ from the RTS (subscript "lib") are calculated directly from the given $r_{\mathrm{eff,lib}}$, which represents the cloud top $r_{\mathrm{eff}}$ of the model cloud. Total cloud optical thickness $\tau_{\mathrm{lib}}$ and $r_{\mathrm{eff,lib}}$ from the libradtran radiative transfer simulations are considered to be the reference values which are

used to compare the retrieval results and the calculated $N$. For consistency the labeling of $N$ for the three methods follows Section 5. An overview of all retrieved and calculated parameters is given in Table 3.

The retrieved cloud optical thickness $\tau_{\mathrm{R}}$ is higher compared to the true value $\tau_{\mathrm{lib}}$ for all cloud cases. The largest difference of $26\%$ are observed for cloud I. With increasing $N_{\mathrm{cld}}$ the absolute and relative differences become smaller. Systematically larger errors are found for the adiabatic clouds. A similar pattern is obtained for $r_{\mathrm{eff,R}}$ which is always up to $2\%$ smaller then

$r_{\mathrm{eff,lib}}$. The sub-adiabatic clouds show the largest differences. The relative error decreases for higher $N_{\mathrm{cld}}$. The systematic underestimation of $r_{\mathrm{eff,R}}$, especially for the sub-adiabatic cases, with respect to $r_{\mathrm{eff,lib}}$ results from the penetration depth of the incident solar radiation into the cloud. For constant $LWP$, clouds with lower $N$ have a lower $\tau$, which reduces the scattering. Therefore, the incident radiation can penetrate deeper into the cloud compared to clouds with higher $N$ and $\tau$ (Platnick, 2000). As a result, $I^{\uparrow}_{\lambda}$ is more influenced by lower cloud layers and the retrieved $r_{\mathrm{eff,R}}$ is systematically smaller than $r_{\mathrm{eff,lib}}$. In this

case, $r_{\mathrm{eff,R}}$ is not representing $r_{\mathrm{eff,lib}}$ at cloud top. The bias of $r_{\mathrm{eff,R}}$ from the $r_{\mathrm{eff,lib}}$ at cloud top feeds back into the retrieval of $\tau_{\mathrm{R}}$ because of the dependence of $\tau$ and $r_{\mathrm{eff}}$ and the non-rectangular shape of the Look-Up-Table. The overall underestimation of retrieved $r_{\mathrm{eff,R}}$, which appears for all passive remote sensing measurements based on reflected solar radiation, generally leads to an overestimation of $N$, which is intensively discussed e.g. by Brenguier et al. (2000) and Grosvenor et al. (2018b, a) and therefore, not repeated here.





Liquid water path $LWP_R$ is calculated with Eq. (9) from the retrieved $\tau_R$ and $r_{eff,R}$ by assuming an adiabatic cloud profile. In all cases, the retrieval overestimates $LWP_R$ by 18% for low $N_{cld}$ up to 27%. The deviation becomes larger for high $N_{cld}$.

The cloud droplet number concentration $N_{A,lib}$ is calculated with method A by using $\tau_{lib}$, $r_{eff,lib}$, and assuming an adiabatic vertical profile with $\Gamma_{ad}$. This provides a reference for $N_{A,R}$ which applies $\tau_R$ and $r_{eff,R}$. By comparing $N_{A,lib}$ and $N_{A,R}$ the

influence of the remote sensing retrieval method (forward simulations and error due to penetration depth) on $N$ for different $N_{cld}$ becomes obvious. In general, $N_{A,lib}$ and $N_{A,R}$ of all clouds are larger compared to $N_{cld}$. Differences between $N_{A,lib}$, $N_{A,R}$ and $N_{cld}$ result from smaller retrieved $r_{eff,R}$ and higher $\tau_{lib}$ compared to $\tau_R$. Another reason is the difference between $\Gamma_{ad}$ used in the model cloud and the assumed $LWP$ parameterization in Eq. (9) which is applied in Eq. (7) to correlate $LWP$ and $\tau$. For all clouds, $N_{A,R}$ is larger then $N_{A,lib}$ and $N_{cld}$, because in Eq. (7) $N$ is dominated by $r_{eff}^{-5/2}$ and less sensitive to

$\tau^{1/2}$. Differences between $N_{A,lib}$ and $N_{A,R}$ vary between 0% and 17%, being largest for cloud I for which the deviation in $r_{eff,lib}$ and $r_{eff,R}$ is largest. The simulations also show that $N_{A,lib}$ and $N_{A,R}$ are largest for the sub-adiabatic cloud cases.

For method B, $N_{B,lib}$ and $N_{B,R}$ are larger then $N_{cld}$ with smaller differences for the reference values of $N_{B,lib}$ and larger differences of $N_{B,R}$ compared to $N_{cld}$. For method B the deviations of $N_{B,lib}$ and $N_{B,R}$ compared to $N_{cld}$ are largest for the sub-adiabatic cloud cases. The systematic overestimation of $N_{B,R}$ for all clouds is due to the lower $r_{eff,R}$. The differences

reduce for increasing $N_{cld}$ because the differences between $r_{eff,R}$ and $r_{eff,lib}$ decrease. This clearly shows that a wrong estimation of $r_{eff}$ influences the calculation of $N$ most significantly, while $\tau$ contributes to a minor part only, independently which method is used. These results allow to conclude that $r_{eff}$ must be retrieved close top. This is possible if the retrieval applies appropriate wavelength in the infrared, where radiation is effectively absorbed within the upper most part of the cloud. Otherwise systematic overestimation of $N$ occurs.

By applying method C the sub-adiabatic nature of the cloud profiles (II, IV, VI) is considered in the estimation of $N$. The calculated $\Gamma_{calc}$ is assumed to be correct and identical to the profile of the constructed clouds, with $f_{ad} = 0.6$ and $\Gamma_{calc} = \Gamma_{ad} \cdot 0.6$, respectively. Therefore, it is obvious, that $N$ calculated from method B and C are also identical for adiabatic clouds. In general, $N_{C,R}$ derived from method C is closer to $N_{cld}$ than $N_{B,R}$. However, for the sub-adiabatic clouds (II, IV, VI) results for methods B and C differ. Cloud droplet number concentration $N_{C,lib}$ is closest to $N$ for all cloud cases and methods. The same pattern

is present for $N_{C,R}$ with the best agreement to $N_{cld}$ compared to method A and B. Deviations in $N_{C,lib}$ and $N_{C,R}$ to $N_{cld}$ are reduced with increasing $N$. This shows, that a correct assumption of $\Gamma_{calc}$, as possible with method C, is crucial for a reliable calculation of $N$ and can compensate biases in $N$ which result from the the sub-adiabatic cloud profile.

## 5.5 Calculation of Retrieval Uncertainty of Cloud Droplet Number Concentration

Cloud droplet number concentrations calculated with Eq. (7), Eq. (10), and Eq. (12) are mainly effected by uncertainties from $\tau$, $LWP$ and especially $r_{eff}$, but also depend on the accuracy of $k$, $f_{ad}$, and $\Gamma_{ad}$. To estimate the uncertainties of retrieved $N$, it is assumed that the errors are normally distributed and independent from each other. In this case the uncertainty of $N_A$ from





**Table 3.** Overview of all six synthetic cloud cases. The predefined cloud liquid water path $LWP$ and droplet number concentration $N$ are denoted with subscript "cld". Cloud properties are calculated from the given cloud profile (subscript "lib") and retrieved from synthetic spectral cloud reflectivities (subscript "R"). Calculated $N$ is listed for all three methods A,B, and C once using the predefined cloud properties "lib" and the retrieval results from "R".

|  | Cloud I adiabatic | Cloud II sub-adiabatic | Cloud III adiabatic | Cloud IV sub-adiabatic | Cloud V adiabatic | Cloud VI sub-adiabatic |
|---|---|---|---|---|---|---|
| $N_{\mathrm{cld}}$ [cm$^{-3}$] | 50 | 50 | 100 | 100 | 200 | 200 |
| $LWP_{\mathrm{cld}}$ [g m$^{-2}$] | 362 | 217 | 362 | 217 | 362 | 217 |
| $\tau_{\mathrm{lib}}$ | 35.6 | 25.5 | 45.2 | 32.3 | 57.3 | 41.0 |
| $r_{\mathrm{eff,lib}}$ [$\mu$m] | 18.8 | 18.8 | 14.9 | 12.6 | 11.8 | 10.0 |
| $\tau_{\mathrm{R}}$ | 37.1 | 25.7 | 46.9 | 32.8 | 59.4 | 41.9 |
| $r_{\mathrm{eff,R}}$ [$\mu$m] | 18.3 | 15.4 | 14.8 | 12.3 | 11.9 | 9.9 |
| $LWP_{\mathrm{R}}$ [g m$^{-2}$] | 452 | 264 | 462 | 270 | 471 | 276 |
| $N_{\mathrm{A,lib}}$ [cm$^{-3}$] | 53 | 69 | 106 | 137 | 215 | 274 |
| $N_{\mathrm{A,R}}$ [cm$^{-3}$] | 58 | 74 | 111 | 145 | 215 | 288 |
| $N_{\mathrm{B,lib}}$ [cm$^{-3}$] | 52 | 68 | 105 | 134 | 211 | 268 |
| $N_{\mathrm{B,R}}$ [cm$^{-3}$] | 57 | 73 | 108 | 143 | 207 | 280 |
| $N_{\mathrm{C,lib}}$ [cm$^{-3}$] | 52 | 53 | 105 | 104 | 211 | 208 |
| $N_{\mathrm{C,R}}$ [cm$^{-3}$] | 57 | 57 | 108 | 111 | 207 | 217 |

Eq. (7) is calculated by:

$$\Delta N = \sqrt{\left(\frac{\partial N}{\partial k}\right)^2 (\Delta k)^2 + \left(\frac{\partial N}{\partial f_{\mathrm{ad}}}\right)^2 (\Delta f_{\mathrm{ad}})^2 + \left(\frac{\partial N}{\partial \Gamma_{\mathrm{add}}}\right)^2 (\Delta \Gamma_{\mathrm{add}})^2 + \left(\frac{\partial N}{\partial \tau}\right)^2 (\Delta \tau)^2 + \left(\frac{\partial N}{\partial r_{\mathrm{eff}}}\right)^2 (\Delta r_{\mathrm{eff}})^2} \tag{14}$$

and analogous for Eq. (10) and Eq. (12). All uncertainties of $N$ presented in the following sections are based on calculation by this approach. The uncertainties of the single parameters assumed in the calculations are summarized below.

5 For method A, B, and C, the uncertainty of $k$ representing the shape of the droplet size distribution is set to $k = 0.8 \pm 0.1$ according to the range of values suggested by Martin et al. (1994) and Pontikis and Hicks (1992).

For methods A and B the degree of adiabiticity $f_{\mathrm{ad}}$ is fixed to one. In that case, no uncertainty in a measurement scene is attributed to $f_{\mathrm{ad}}$. For method C, the uncertainty of $f_{\mathrm{calc}}$ is determined by the uncertainty of $h_{\mathrm{top}}$, $h_{\mathrm{base}}$ and retrieved $LWP$ following (Eq. 11). Cloud top height from WALES is determined with an accuracy of $\Delta h_{\mathrm{top}} = \pm 20\,\mathrm{m}$. The cloud base height

10 is derived from single dropsondes and therefore prone to horizontal variability of $T$, $p$ and $T_{\mathrm{d}}$. Based on an analysis of different dropsondes in close vicinity, an uncertainty of $h_{\mathrm{LCL}} = 660\,\mathrm{m} \pm 35\,\mathrm{m}$ is assumed. The evaluation of all dropsondes show that the thermodynamic conditions in the selected area stayed constant ($\Delta T < 2\,\mathrm{K}$ and $\Delta p < 4\,\mathrm{hPa}$) during the flight time with $h_{\mathrm{top}} \approx 1800\,\mathrm{m}$, $T_{\mathrm{top}} = 20.2\,^{\circ}\mathrm{C}$, and $p_{\mathrm{top}} = 820\,\mathrm{hPa}$. The accuracy of the deployed Vaisala dropsondes RD94 is reported to be





within $\Delta T = \pm 0.2\,\mathrm{K}$ and $\Delta p = \pm 0.4\,\mathrm{hPa}$. Uncertainties of $N_{\mathrm{C}}$ caused by errors in $\Gamma_{\mathrm{ad}}$ are therefore negligible compared to the influence of $\tau$ and $r_{\mathrm{eff}}$.

The adiabatic increase of $LWC$ with height calculated from the Clausius-Clapeyron-Equation depends mostly on cloud top temperature $T_{\mathrm{top}}$ and to a lower degree on cloud top pressure $p_{\mathrm{top}}$. Therefore $\Gamma_{\mathrm{ad}}$ depends on $T_{\mathrm{top}}$ and $p_{\mathrm{top}}$, too. The cloud droplet number concentration is mostly effected by the assumed $T_{\mathrm{top}}$ whereby $p_{\mathrm{top}}$ is only of minor contribution. Despite that, the cloud top pressure more strongly affects warm than cold clouds (Grosvenor et al., 2018b). For the uncertainty calculation, a temperature difference of 2 K is considered, which changes $\Gamma_{\mathrm{ad}}$ by $\pm 0.1 \cdot 10^{-3}\,\mathrm{g\,m^{-3}\,m^{-1}}$ for the reference value of $2.5 \cdot 10^{-3}\,\mathrm{g\,m^{-3}\,m^{-1}}$.

The uncertainty of the retrieval of $\tau$ and $r_{\mathrm{eff,A}}$ result from the measurements uncertainties of SMART which are described in Sec. 3.1. For typical trade wind cumuli uncertainties of $\pm 0.1$ for $\tau$ and $\pm 1.1\,\mu\mathrm{m}$ for $r_{\mathrm{eff,A}}$ are assumed.

Retrieval uncertainties of $LWP$ from HAMP are in the range of $\pm 30\,\mathrm{g\,m^{-3}}$ and mainly stem from the non-ambiguity of the retrieval problem and instrumental effects described in Section 3.2. Common microwave radiometer retrievals of $LWP$ from satellite observations above $180\,\mathrm{g\,m^{-3}}$ are error-prone because of their large footprint. With the smaller footprint of HAMP the value of $180\,\mathrm{g\,m^{-3}}$ is only for rough orientation and the actual accuracy of $LWP$ is a function of $LWP$ itself and has to be estimated for each instrument. As a result, uncertainties in $LWP$ and from within precipitating cloud cases can be even higher then indicated by the dotted lines (Seethala and Horvath, 2010), which results in a higher uncertainty in retrieved $N_{\mathrm{B}}$ ad $N_{\mathrm{C}}$.

Retrievals of $r_{\mathrm{eff,B}}$ from combined measurements of SMART and HAMP are slightly more prone to the uncertainty of the $LWP_{\mathrm{B}}$ measurements and lead to uncertainties of $r_{\mathrm{eff,B}}$ of up to $\pm 1.5\,\mu\mathrm{m}$. This is a higher uncertainty for $r_{\mathrm{eff}}$ than estimated for Method A. However, the uncertainty of $N$ with respect to $r_{\mathrm{eff}}$ is lower as the sensitivity of $N_{\mathrm{B}}$ with respect to $r_{\mathrm{eff,B}}$ is lower in Eq. (10) compared to Eq. (7). Additionally sun-glint or 3D radiative effects are omitted.

For the exemplary ideal adiabatic case discussed above, the total uncertainties of the three methods are for $\Delta N_{\mathrm{A}} = \pm 7.1\,\mathrm{cm^{-3}}$, $\Delta N_{\mathrm{B}} = \pm 14.1\,\mathrm{cm^{-3}}$ and $\Delta N_{\mathrm{C}} = \pm 15.1\,\mathrm{cm^{-3}}$ in the optimal adiabatic case. In case of sub-adiabatic clouds, the uncertainties of method A and B increase due to the assumption of adiabaticity. The additional error in $N$ results from the increased variability in $f_{\mathrm{ad}}$. Additionally the retrieved $r_{\mathrm{eff}}$ is biased to larger values because of the increased penetration depth due to lower optical thickness. Uncertainties fur sub-adiabatic clouds increase to $\Delta N_{\mathrm{A}} = \pm 10.9\,\mathrm{cm^{-3}}$, $\Delta N_{\mathrm{B}} = \pm 23.2\,\mathrm{cm^{-3}}$ whereby the uncertainty $\Delta N_{\mathrm{C}}$ remains constant.

From the error estimation of the retrieval of $N$ it can be concluded that uncertainties in $r_{\mathrm{eff}}$, $LWP$, and $\mathrm{d}z$ have to be minimized as they influence the retrieval the most. Determination of $h_{\mathrm{base}}$, either from the dropsondes or the radar, and resulting $\mathrm{d}z$ have to be accurate within at least $\pm 60\mathrm{m}$.

## 6 Results

The retrieval of $N$ is applied to two measurement cases observed during NARVAL-II. Figure 2 shows the flight track of Research Flight 06 (RF 06) from 19 August 2016 and the flight section (19:24 to 19:39 UTC) of the track for which the remote sensing measurements are analyzed. The satellite image represents the cloud situation at 19:30 UTC. The presence of intense





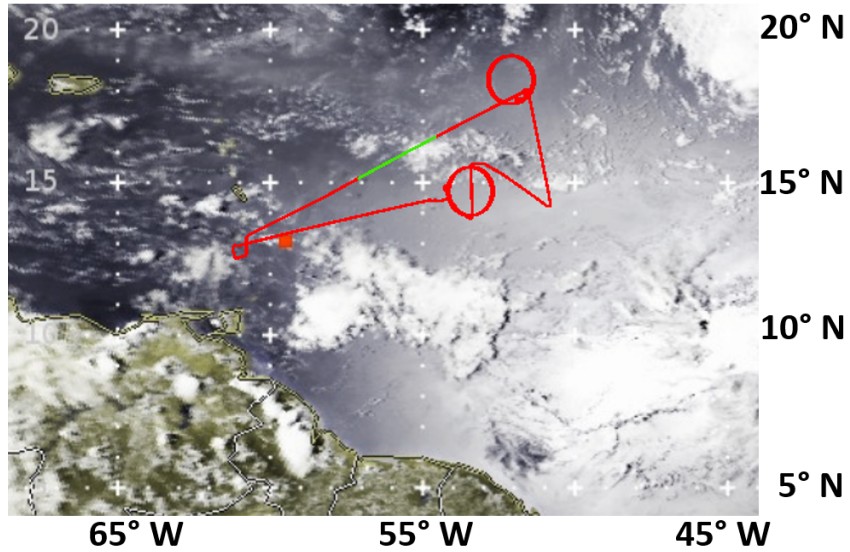

**Figure 2.** Flighttrack of HALO (red) from RF 06 (19. August 2016) plotted on a METEOSAT satellite composite image from 19:30 UTC. The section for which the remote sensing measurements are analyzed (19:24 UTC to 19:39 UTC ) crosses a region with aggregated trade wind cumulus and is plotted in green.

sun-glint is visible, which enhances the reflected radiance $I_\lambda^\uparrow$ and influences the cloud detection (low contrast) and the retrieval of $\tau$ and $r_{\mathrm{eff,A}}$. The analyzed time period is divided into two parts, cloud case #1 and cloud case #2. The north eastern part of the flight track (19:29-19:32 UTC) was dominated by aggregated trade wind cumulus, whereby in the south-western part (19:32-19:36 UTC) shallow cumulus were present. The general weather situation was characterized by moderate convection

5    with low cloud top altitudes. Locally more dense cloud fields formed, at about $10° \mathrm{N}$ and $16° \mathrm{N}$ at $55° \mathrm{W}$.

Time series of measured and retrieved parameters of both cloud cases are shown in Fig. 3 and Fig. 4. The three methods to calculate $N$ assume, that there is no precipitation. As the radar sensitivity is limited, measured $Z$ is most sensitive to large cloud droplets and it can not be guaranteed that drizzle is completely excluded. In these cases the influence of precipitation is assumed to be negligible. Flight sections which are flagged for precipitation are highlighted by the grey boxes. At the top of

10    Fig. 3 and 4 the cloud mask (blue) and the homogeneity cloud flag HCF (yellow) are indicated. Images of RGB composites by specMACS are given in the lower part of the plots to illustrate the visual cloud characteristics. Data gaps are due to cloud free pixel.





### 6.1 Cloud Case #1

Case #1 represents a single layer stratiform cloud without any convective areas which is an ideal test case for the retrieval of $N$. The cloud optical thickness $\tau$ shown in Fig. 3a is generally low and ranges between 0 and 2 at the beginning of the section, while $\tau$ increases up to 6 with time. The uncertainty of $\tau$ is estimated to be $\pm 0.1$. The effective radius $r_{\mathrm{eff,A}}$ (panel b, black line)

ranges between 9.6 $\mu$m and 26.3 $\mu$m with an uncertainty of $\pm 1.0\,\mu$m, while $r_{\mathrm{eff,B}}$ is between 8.3 $\mu$m and 30 $\mu$m retrieved with a slightly higher uncertainty of $\pm 1.5\,\mu$m. For the first cloud part, the liquid water path obtained from SMART $LWP_{\mathrm{A}}$ (panel c) is calculated with Eq. (1) using retrieved $\tau$ and $r_{\mathrm{eff,A}}$. For the first part of the cloud $LWP_{\mathrm{A}}$ is slightly lower than the $LWP_{\mathrm{B}}$ measured by the microwave profiler, while with increasing $\tau$ the agreement between both $LWP$ improves. Vertical profiles of $Z$ shown in Fig. 3g are below the detection threshold except for four cloud patches. This indicates, that no precipitation was

detected, whereby slight drizzle can not be excluded. Cloud base height is estimated from dropsondes to be around 1500 m, while $h_{\mathrm{top}}$ is determined by WALES. The resulting cloud geometric thickness $\mathrm{d}z$ (Fig. 3d) varies between 100 m and 420 m. Cloud adiabaticity $f_{\mathrm{calc}}$ (Fig. 3e) is mostly below 0.5 indicating a considerable sub-adiabatic cloud. Calculated $N_{\mathrm{A}}$ and $N_{\mathrm{B}}$ are shown in Fig. 3f and range between $5\,\mathrm{cm}^{-3}$ and $40\,\mathrm{cm}^{-3}$ which results from the low low $\tau$, $LWP_{\mathrm{B}}$, large $r_{\mathrm{eff,A}}$, and $r_{\mathrm{eff,B}}$. The cloud droplet number concentration $N_{\mathrm{A}}$ shows a peak around 19:34:30 UTC and $N_{\mathrm{A}}$ at 19:35:00 UTC. Cloud droplet

number concentration $N_{\mathrm{C}}$ derived by method C is lower than $N_{\mathrm{A}}$ and $N_{\mathrm{B}}$ and does show a reduced variability compared to $N_{\mathrm{A}}$ and $N_{\mathrm{B}}$. However, the uncertainty of all $N$ is about $\pm 15\,\mathrm{cm}^{-1}$. While in the fist part of cloud case #1 the differences in $N$ are large, there is a good agreement between all three methods in the second part where all results are inside the uncertainty range of each method. Mean values of measured and retrieved parameters for cloud case #1 are listed in Table 4.

### 6.2 Cloud Case #2

The second case represents a more heterogeneous single-layer cloud observed between 19:29 UTC and 19:32 UTC and is shown in Fig. 4. This cloud is in a later state of development and shows moderate convection with slight precipitation. In these areas (highlighted in grey), the criteria for cloud homogeneity is not fulfilled. Despite that and the slight precipitation, calculation of $N$ is performed, knowing that the retrieval of $N$ using method A and B are prone to errors under this circumstances.

These results are used to evaluate the improvement of retrieved $N$ by method C which accounts for cloud geometry and subadiabicity. The application of method C provides reasonable $N$ by accounting for cloud-geometry and sub-adiabaticity. By comparing convective and non-convective areas of this cloud case #2, the limitations and advantages of the three methods are investigated. Mean values of the measured and retrieved parameters from the three different methods separated for nonprecipitation and precipitation are summarized in Table 4.

For the non-precipitating and homogeneous part of cloud case #2, $\tau$ does not exceed a value of 30 and $r_{\mathrm{eff,A}}$ and $r_{\mathrm{eff,B}}$ range between 18 $\mu$m and 40 $\mu$m (Fig. 4a, b). The uncertainty of all measured and retrieved parameters, is in a similar range as calculated for cloud case #1. Retrieved $LWP$ from SMART and HAMP (Fig. 4c) agrees within the uncertainty range of HAMP for most parts of the homogeneous cloud sections. Larger differences appear around 19:29:30 UTC where $LWP_{\mathrm{A}}$ is larger than





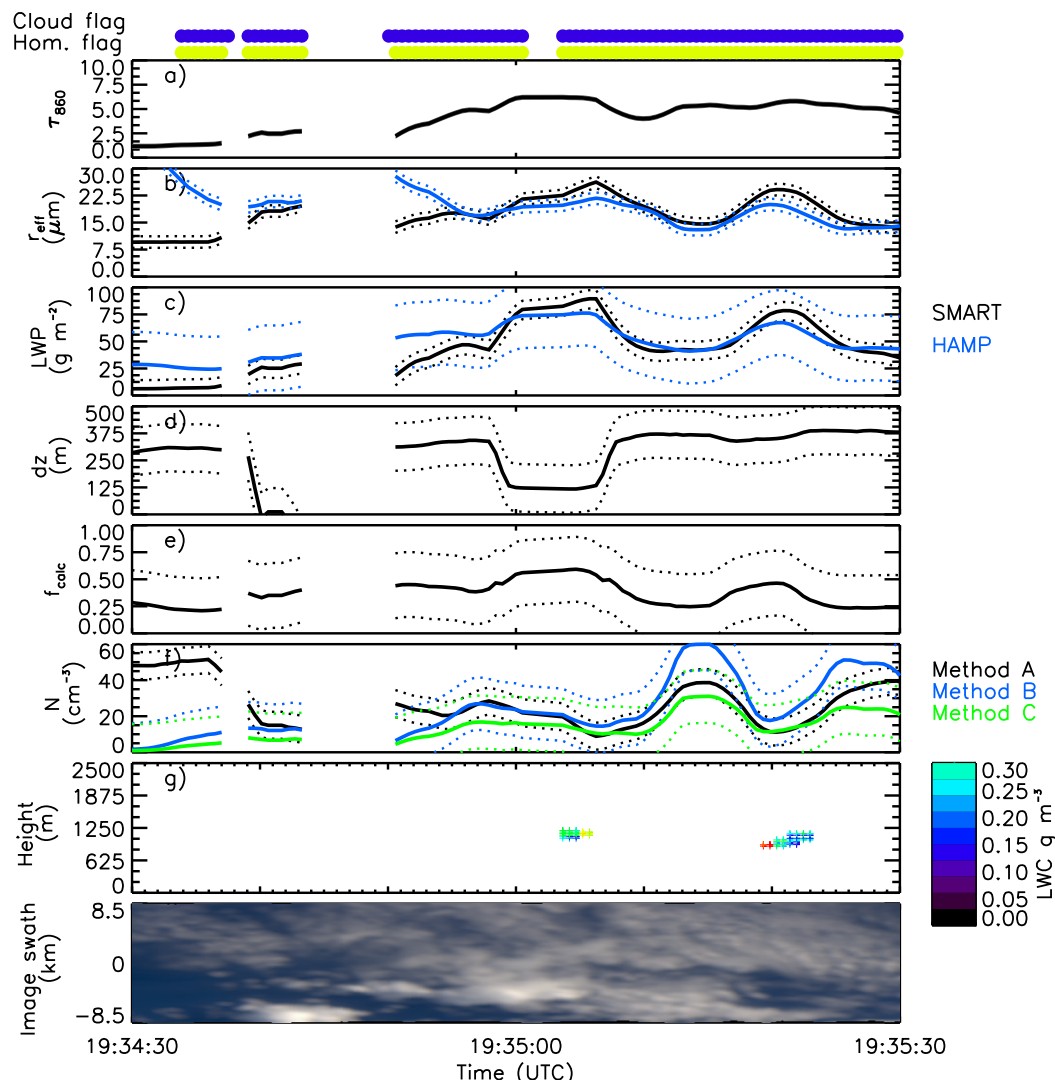

**Figure 3.** Time series of measured and retrieved cloud properties of cloud case #1 from 19:34:30 to 19:35:30 UTC of RF06. Cloud droplet number concentration $N$ is shown for all three methods A, B, and C. Uncertainty ranges of the individual parameters are indicated by dotted lines. At the top, the cloud mask (blue) and the homogeniety cloud flag (HCF) (yellow) derived by SMART are indicated.





$LWP_\text{B}$. For method C, cloud geometrical thickness is calculated from a combination of HAMP and WALES. Radar reflectivity is above the detection threshold and allows to determine vertical profiles of the $LWC$ and $h_\text{base}$ with an average value of $\approx 900$ m where no precipitation is present. Cloud top height is determined from WALES and ranges between 200 m and 1000 m for the non-precipitating regions. This results in a highly variable $f_\text{calc}$, which varies between strongly varies between

0.05 and 1.0.

Cloud droplet number concentration from method A and B calculated for cloud case #2 are generally low (see also Table 4) mostly ranging between $20\,\text{cm}^{-3}$ and $40\,\text{cm}^{-3}$. Together with large $r_\text{eff,A}$ and $r_\text{eff,B}$ these values indicate typical pristine maritime clouds. An exception is observed around 19:29:30 UTC where $N$ peaks up to $120\,\text{cm}^{-3}$ for all three methods mostly resulting from a decrease of $r_\text{eff,A}$ and an increase of $\tau$. The decrease of $r_\text{eff}$ might result from 3D-radaitive effects at the cloud

edge overestimating the cloud particle size and can have biased the retrieval of $N$.

In the areas marked with precipitation, retrieved $\tau$, $LWP_\text{A}$, and $LWP_\text{B}$ are higher compared to the precipitation free regions while $r_\text{eff,A}$ and $r_\text{eff,B}$ are in the same range as for the non-precipitating areas. In contrast to the homogeneous parts of the cloud, the convective regions show stronger horizontal heterogeneity in all parameters. The optical thickness reaches up to 40 and $r_\text{reff,A}$ ranges from 20 $\mu$m to 38 $\mu$m. In these areas the $LWP_\text{B}$ from HAMP exceeds $270\,\text{g}\,\text{m}^{-3}$ and shows a maximum

value up to $500\,\text{g}\,\text{m}^{-3}$. Liquid water path from SMART is in the same range of $LWP_\text{B}$ except for the first precipitation section (19:30:30 UTC) where $LWP_\text{B}$ is lower than $LWP_\text{A}$. For the precipitating regions the cloud base height $h_\text{base}$ is assumed to be at the same level as determined for the non-precipitating regions as precipitation makes the cloud base invisible for the radar. The cloud geometric thickness $\text{d}z$ is slightly higher for the connective regions and ranges between 800 m and 1300 m. The calculated adiabaticity $f_\text{calc}$ is lower than 0.5 for the majority of the measurement and shows that most parts of the cloud are

sub-adiabatic. For the precipitation regions calculated $N$ are between $10\,\text{cm}^{-3}$ and $90\,\text{cm}^{-3}$ with the highest concentrations for method B, followed by method A and the lowest $N$ for method C. In the areas with precipitation, $N$ shows a systematic higher variability which is observed by all three methods and likely caused by the variability of $r_\text{eff}$ retrieved from SMART. One reason for this variability is the relation of $r_\text{vol}$ to $r_\text{reff}$ which is assumed to be (i) constant in the retrieval of $r_\text{A}$ and $r_\text{B}$ and (ii) significantly influenced by formation of precipitation. Therefore, calculated $N$ by all three methods are highly prone

to errors for precipitating clouds. The variability of $N$ might also be caused by intense turbulent mixing processes within the cloud. Therefore, it is suggested to filter areas with stronger convection, precipitation, and heterogeneous scenes and analyze the retrieved $N$ with special care.

### 6.3    Statistical Analysis of Liquid Water Path, Effective Radius and Cloud Droplet Number Concentration

Statistics of retrieved cloud properties are analyzed for measurements between 19:24:00 UTC and 19:39:00 UTC only, where

the HCF indicates homogeneous clouds and uncertainties of the retrieved cloud parameters is low. An extension of the analysis to other flights is not possible yet, because the reliable application of the retrieval of $N$ requires careful data selection and good quality data of all individual instruments. However, in total still 700 individual measurements are included which represents a cloud field of 77 km length. The clouds were separated into precipitating (p) and non-precipitating (np) pixel. Mean for each measurement are given in Table 4.





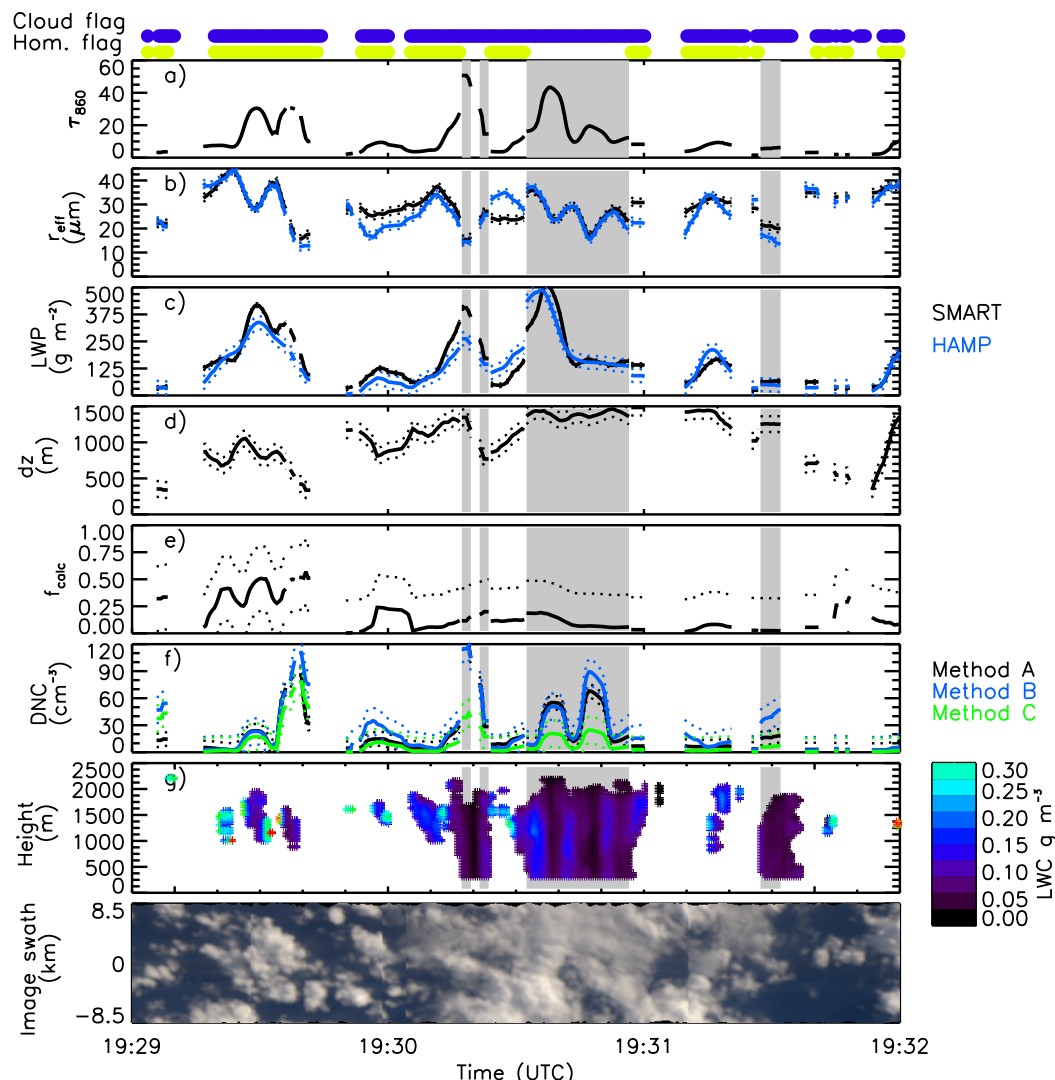

**Figure 4.** Time series of measured and retrieved cloud properties of cloud case #1 from 19:29 to 19:32 UTC of RF06. Cloud droplet number concentration $N$ is shown for all three methods A, B, and C. Uncertainty ranges of the individual parameters are indicated by dotted lines. At the top, the cloud mask (blue) and the homogeniety cloud flag (HCF) (yellow) derived by SMART are indicated.





**Table 4.** Mean values of cloud properties of cloud cases #1 and #2.

| parameter | Cloud case #1 | Cloud case #2 (np) | Cloud case #2 (p) |
|---|---|---|---|
| $\tau$ | 4.3 | 3.5 | 11.3 |
| $r_{\text{eff,A}}$ [$\mu$m] | 17.1 | 30.4 | 24.9 |
| $r_{\text{eff,B}}$ [$\mu$m] | 19.2 | 29.1 | 23.4 |
| $LWP_{\text{A}}$ [$\text{g m}^{-2}$] | 45 | 135 | 226 |
| $LWP_{\text{B}}$ [$\text{g m}^{-2}$] | 50 | 120 | 210 |
| d$z$ [m] | 315 | 959 | 1315 |
| $N_A$ [$\text{cm}^{-3}$] | 27 | 17 | 47 |
| $N_B$ [$\text{cm}^{-3}$] | 26 | 25 | 53 |
| $N_C$ [$\text{cm}^{-3}$] | 19 | 13 | 40 |

Figure 5 compares measurements of $LWP_{\text{A}}$ and $LWP_{\text{B}}$. The data is separated for different $r_{\text{eff,A}}$ split into bins of 5 $\mu$m size. For the selected time period, $LWP_{\text{A}}$ agrees with $LWP_{\text{B}}$ within the uncertainty range of HAMP of $\pm 30 \, \text{g m}^{-2}$ indicated by the grey error bars. The differences of $LWP_{\text{A}}$ and $LWP_{\text{B}}$ show a larger variability for clouds with large $r_{\text{eff,A}}$ than for clouds with small $r_{\text{eff,A}}$. For larger cloud droplets, the retrieval uncertainty of $\tau$ and $r_{\text{eff,A}}$ increases and, therefore, also $LWP_{\text{A}}$ de-
rived from SMART. Additionally, SMART has a higher sensitivity to droplets at cloud top and the FOV of HAMP is slightly larger compared to SMART what can explain some of the observed variability. Slightly different viewing directions have to be considered too. While for SMART the $LWP_{\text{A}}$ is calculated assuming an adiabatic profile with the retrieved $r_{\text{eff,A}}$ representing cloud top, HAMP obtains an integrated measure of $LWP$ where all cloud layers are more homogeneously weighted and no assumption on the cloud profiles is required. Therefore, a difference between $LWP_{\text{A}}$ and $LWP_{\text{B}}$ indicates that the observed
clouds are non-adiabatic. For $LWP_{\text{A}} > LWP_{\text{B}}$ less liquid water is at cloud base than predicted by adiabatic theory and clouds are sub-adiabatic. For $LWP_{\text{A}} < LWP_{\text{B}}$ liquid water at cloud top is reduced, likely by precipitation.

In Fig. 7 the normalized PDF of $r_{\text{eff,A}}$ retrieved from SMART only (method A) and $r_{\text{eff,B}}$ retrieved synergistically from SMART and HAMP from (method B) is presented separately for precipitating and non-precipitating clouds. The mean value for non-precipitating clouds is around $\overline{r}_{\text{eff,A,np}} = 23.2 \, \mu$m and the median is at $r_{\text{eff,A,np,med}} = 21.1 \, \mu$m. This droplet size
range agrees with in-situ measurements of pristine trade wind cumulus by Siebert et al. (2013) and remote sensing measurements by Werner et al. (2014) in the same geographic region. The distribution shows a bi-modal structure with a first mode around 15 $\mu$m and a second mode around 32 $\mu$m. The PDF of $r_{\text{eff,A}}$ for precipitation situations shows a similar structure being shifted towards larger $r_{\text{eff,A}}$ with values of $\overline{r}_{\text{eff,A,p}} = 25.1 \, \mu$m and $r_{\text{eff,A,p,med}} = 24.5 \, \mu$m. The fist mode is at 21 $\mu$m and the second mode is at 36 $\mu$m. The PDF's of $r_{\text{eff,B}}$ for the np clouds are shifted to larger values by approximately 3 $\mu$m additionally
showing a third mode around 38 $\mu$m. In contrast, the PDF for the p clouds is shifted to lower values by up to 8 $\mu$m and showing only the bi-modal structure with peaks around 15 $\mu$m and 33 $\mu$m.

Figures 8a and b show normal normalized PDF of the calculated $N$ for non-precipitating (a) and precipitating regions (b) of the





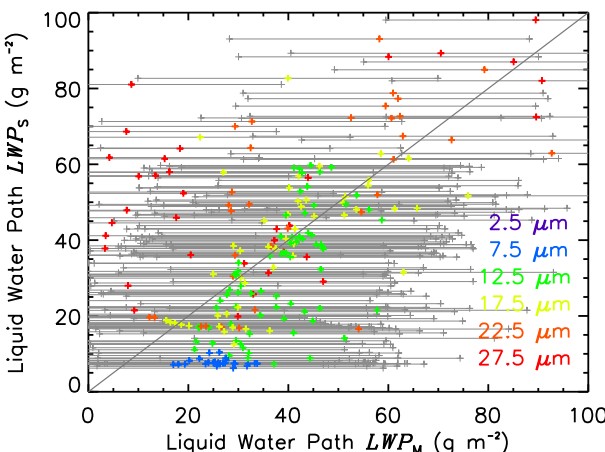

**Figure 5.** Comparison of liquid water path $LWP_B$ from HAMP microwave radiometer and $LWP_A$ calculated from $\tau$ and $r_{eff,A}$ retrieved by SMART. The color code indicates different ranges of $r_{eff,A}$. HAMP uncertainties of $LWP$ ($\pm 30\,\mathrm{g\,m^{-2}}$) are indicated by grey errors bars.

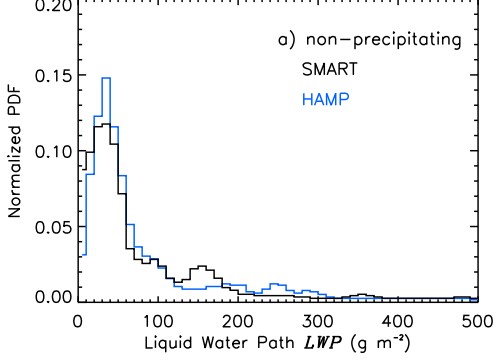

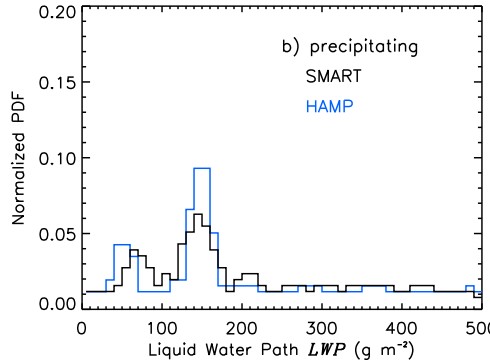

**Figure 6.** Normalized probability density function (PDF) of measured and calculated liquid water path $LWP$ from HAMP (blue) and SMART (black). Distributions are filtered for non-precipitating a) and precipitating b) clouds.



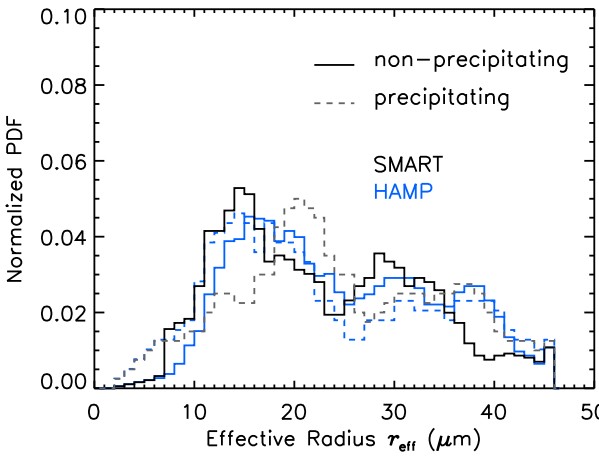

**Figure 7.** Normalized probability density function (PDF) of the effective radius $r_{\mathrm{eff,A}}$ retrieved by using the ratio of 1645 nm to 1050 nm in black and $r_{\mathrm{eff,B}}$ from the combined spectrometer-microwave retrieval in blue. Distributions are filtered for non-precipitating (solid line) and precipitating (dashed line) clouds.

selected flight-leg from all three methods A, B and C. For non-precipitating clouds (panel 8a) the distribution of $N_{\mathrm{A}}$ peaks at $\approx 30\,\mathrm{cm}^{-3}$ with a steep decrease towards a concentration of $\approx 100\,\mathrm{cm}^{-3}$. The first local maximum of the $N_{\mathrm{B}}$ distribution is at $\approx 30\,\mathrm{cm}^{-3}$ slowly decreasing for larger $N$. Only a slight difference between $N_{\mathrm{A}}$ and $N_{\mathrm{B}}$ is present for higher $N_{\mathrm{A}}$. This can be explained by the slightly higher values of $LWP_{\mathrm{A}}$ retrieved by SMART compared to $LWP_{\mathrm{B}}$ observed by HAMP. The PDFs of

$N_{\mathrm{A}}$ and $N_{\mathrm{B}}$ show reasonable results for pristine, maritime clouds with relative large $r_{\mathrm{eff,A}}$ and according low $N$ from method A and B. Cloud droplet number concentration from method C are significantly lower as a result of the considered adiabaticity of the individual clouds.

Measurements affected by precipitation compared to Fig. 8a show almost the same distribution with a shift to larger $N$ for all three calculation methods, especially for method C. Filtering for precipitating clouds the statistic might be biased by only

considering further developed clouds in which precipitation formation changes and broadens the droplet size distribution. This leads to differences in the means of $r_{\mathrm{vol}}$ and $r_{\mathrm{eff}}$, influencing the $k$-parameter which is assumed to be 0.8 in the $N$ calculation. Retrieving $k$ by passive remote sensing is not possible yet (Wood, 2006).

Figure 9 shows the cloud top reflectivity $\mathcal{R}_{532}$ measured by SMART at 532 nm as a function of $N_{\mathrm{B}}$ retrieved from combined SMART and HAMP measurements. Only measurements of the flight leg where no precipitation was observed are presented.

The data is binned for two different $LWP$. Figure 9a shows clouds with $LWP$ between $0 - 50\,\mathrm{g\,m}^{-2}$ and Fig. 9b shows clouds in the range between $50 - 100\,\mathrm{g\,m}^{-2}$. Colors represent $r_{\mathrm{eff,B}}$ binned from 5 to 30 $\mu$m in 5 $\mu$m steps (label in Fig. 9 refers to the mean bin value). Using $\mathcal{R}_{532}$ as a measure for the reflectivity of the cloud, the sensitivity of $\mathcal{R}_{532}$ changes of $N$ is comparable to the model based sensitivity study in Section 2. Therefore, in Fig. 9 radiative transfer simulations of theoretical





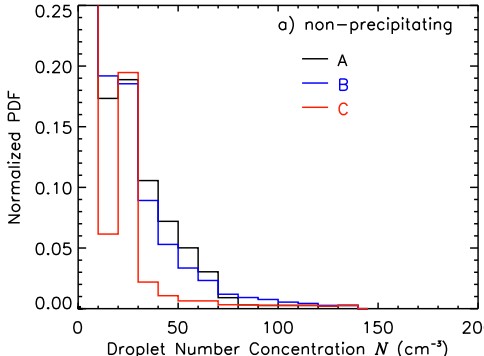 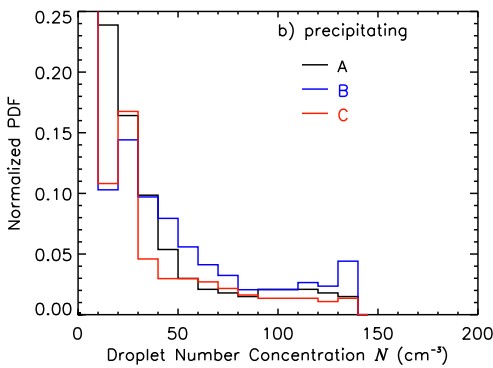

**Figure 8.** Normalized probability density function of the cloud droplet number concentration $N$ for the selected flight path using method A, B, and C. Distributions are filtered for non-precipitating a) and precipitating b) clouds.

$\mathcal{R}_{532,sim}$ for clouds of the same $LWP$ are added by the red line. For the thin clouds in Fig. 9a the measured $\mathcal{R}_{532}$ shows a clear increase for higher $N_B$ over the entire measurement range. This correlation is less pronounced for the thicker clouds in Fig. 9b due to a reduced range of $\mathcal{R}_{532}$ and $N$, that may not cover the entire natural variability. However, for both cloud sub-samples, the measurements follow the theoretical line given by the simulations only that the measured $\mathcal{R}_{532}$ are too low

5  or retrieved $N$ to high. Both might be attributed to measurement biases either the radiometric calibration of SMART or the retrieved $LWP_B$ and $r_{eff,B}$ which feed the calculation of $N_B$. Additionally, the homogeneous assumption of cloud properties applied in the RTS can lead to an overestimation of $\mathcal{R}_{532,sim}$ compared to the measurements. The subdivision of data for different $r_{eff,B}$ shows that clouds in an early developing state with low $LWP_B$ (Fig. 9a) are dominated by smaller cloud droplets up to $r_{eff,B} = 17.5\,\mu\text{m}$ whereby clouds in a later development state with higher $LWP_B$ (Fig. 9b) are dominated by cloud droplets

10  larger than $r_{eff,B} = 17.5\,\mu\text{m}$.





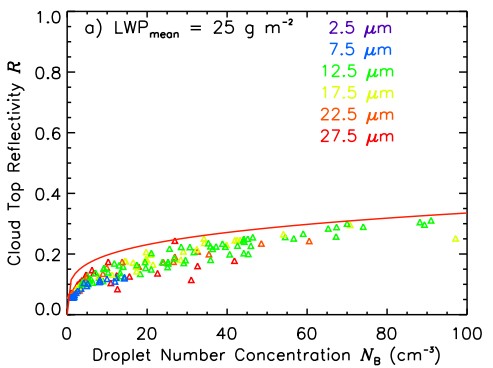 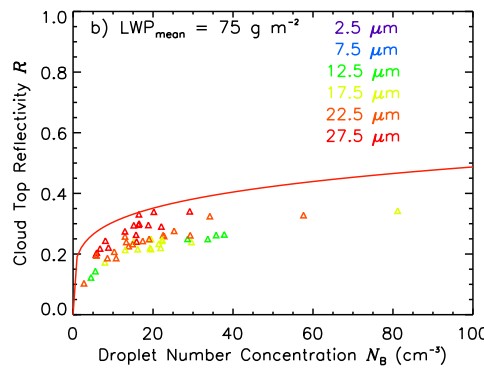

**Figure 9.** Cloud top reflectivity $\mathcal{R}_{532}$ as a function of cloud droplet number concentration $N_B$ for homogeneous, non-precipitating clouds of different liquid water path $LWP_B$ (panel a: $0 - 50\,\mathrm{g\,m^{-2}}$, panel b: $50 - 100\,\mathrm{g\,m^{-2}}$). The droplet effective radius $r_{\mathrm{eff}}$ of each measurement is indicated by the color code. The red line represents simulated reflectivity $\mathcal{R}_{532}$ from radiative transfer calculations for clouds with same $LWP$.

## 7  Conclusions

Trade wind cumulus are an ubiquitous cloud type in the tropics, and it influences the Earth radiation budget significantly. Despite that, these clouds are not well represented in numerical weather prediction (NWP) and global climate models (GCMs), causing considerable uncertainties in the radiative calculations. It is well known, e.g., Platnick and Twomey (1994), that the

albedo $\alpha$ of clouds with low cloud droplet number concentration $N$ and low liquid water path $LWP$, such as trade wind cumulus, respond very sensitive to changes of $N$. In order to obtain improved parameterizations and global distributions of $N$, several methods, including active and passive remote sensing from ground and satellite are developed, but no operational products are available yet. Only a limited number of field campaigns with in-situ measurements of selected cloud cases exist. As a result, the natural variability of trade wind cumulus is poorly covered by appropriate measurements.

In this paper, it is shown that shallow trade wind cumulus with $LWP$ below $200\,\mathrm{g\,m^{-2}}$ and $N$ below $100\,\mathrm{cm^{-3}}$ are very sensitive to changes in $N$ and, therefore, their variable influence on the Earth radiation budget is high. In case of a $LWP$ of $75\,\mathrm{g\,m^{-3}}$ and an increase of $N$ from $50\,\mathrm{cm^{-3}}$ to $100\,\mathrm{cm^{-3}}$, the cloud top albedo can increase by 0.1. As a result, the interaction between cloud top albedo $\alpha$, $N$, $\tau$, $r_{\mathrm{eff}}$ and different thermodynamic conditions (e.g., varying $LWP$) has to be investigated systematically.

Applying the common satellite retrieval techniques of $N$ to measurements conducted with a high flying aircraft, such as the High Altitude and Long Range Research Aircraft (HALO), shows the potential of combined airborne passive and active remote sensing instruments. Using aircraft instead of satellite platforms allows to investigate specific cloud types under selected atmospheric conditions, eg.g., $T_{\mathrm{top}}$, $p_{\approx \bowtie}$, $LWP$.



**Table 5.** List of symbols, longnames and related units.

| Symbol | Longname | Unit |
|---|---|---|
| $\alpha$ | Cloud top albedo | - |
| $D$ | Cloud droplet diameter | m |
| $\mathrm{d}z$ | Cloud geometric thickness | m |
| $f_{\mathrm{ad}}$ | Degree of adiabaticity | - |
| $F_\lambda^\uparrow$ | Spectral upward radiance | $\mathrm{W\,m^{-2}\,nm^{-1}}$ |
| $F_\lambda^\downarrow$ | Spectral downward radiance | $\mathrm{W\,m^{-2}\,nm^{-1}}$ |
| $\Gamma_{\mathrm{ad}}$ | Adiabatic rate of liquid water content | $\mathrm{kg\,m^{-3}\,m^{-1}}$ |
| $\Gamma_{\mathrm{calc}}$ | Calculated rate of liquid water content | $\mathrm{kg\,m^{-3}\,m^{-1}}$ |
| $h_{\mathrm{base}}$ | Cloud base height | m |
| $h_{\mathrm{LCL}}$ | Lifting condensation level | m |
| $h_{\mathrm{top}}$ | Cloud top height | m |
| $I_{\mathrm{cr}}^\uparrow$ | Spectral upward irradiance threshold | $\mathrm{W\,m^{-2}\,nm^{-1}\,sr^{-1}}$ |
| $I_\lambda^\uparrow$ | Spectral upward irradiance | $\mathrm{W\,m^{-2}\,nm^{-1}\,sr^{-1}}$ |
| $I_{\lambda,\mathrm{syn}}^\uparrow$ | Spectral upward irradiance (simulated) | $\mathrm{W\,m^{-2}\,nm^{-1}\,sr^{-1}}$ |
| $k$ | $k$-parameter | - |
| $l_{\mathrm{cld}}$ | Cloud length | m |
| $LWC$ | Liquid water content | $\mathrm{kg\,m^{-3}}$ |
| $LWP$ | Liquid water path | $\mathrm{kg\,m^{-2}}$ |
| $LWP_{\mathrm{A}}$ | Liquid water path from SMART | $\mathrm{kg\,m^{-2}}$ |
| $LWP_{\mathrm{B}}$ | Liquid water path from HAMP | $\mathrm{kg\,m^{-2}}$ |
| $N$ | Cloud droplet number concentration | $\mathrm{cm^{-3}}$ |
| $N_{\mathrm{cld}}$ | Cloud droplet number concentration of simulated clouds | $\mathrm{cm^{-3}}$ |
| $p_{\mathrm{top}}$ | Cloud top pressure | Pa |
| $Q$ | Extinction coefficient | - |
| $\mathcal{R}$ | Cloud top reflectivity | - |
| $p_{\mathrm{Top}}$ | Cloud top pressure | Pa |
| $\rho_{\mathrm{w}}$ | Density of liquid water | $\mathrm{kg\,m^{-3}}$ |
| $r_{\mathrm{eff}}$ | Effective radius | $\mu\mathrm{m}$ |
| $r_{\mathrm{eff,A}}$ | Effective radius from SMART | $\mu\mathrm{m}$ |
| $r_{\mathrm{eff,B}}$ | Effective radius from SMART & HAMP | $\mu\mathrm{m}$ |
| $r_{\mathrm{vol}}$ | Volumetric radius | $\mu\mathrm{m}$ |



| Symbol | Longname | Unit |
|---|---|---|
| $\tau$ | Cloud optical thickness from SMART | - |
| $\tau_{\text{lib}}$ | Cloud optical thickness from libradtran | - |
| $T$ | Temperature | $^\circ$C |
| $T_{\text{d}}$ | Dew-point temperature | $^\circ$C |
| $T_{\text{Top}}$ | Cloud top temperature | $^\circ$C |
| $t_{\text{int}}$ | Integration time of spectrometer | s |
| $v_{\text{aircraft}}$ | Aircraft velocity | $\text{m s}^{-1}$ |
| $\vartheta$ | Solar zenith angle | $^\circ$ |
| $Z$ | Radar reflectivity | dBz |
| $\zeta$ | Cloud top albedo sensitivity | $\text{cm}^3$ |

During the second campaign of the Next Generation Remote Sensing for Validation Studies (NARVAL-II), HALO was equipped with a set of passive and active remote sensing instruments. The Spectral Modular Airborne Radiation measurement sysTem (SMART) measured upward and downward spectral irradiance $F_\lambda$ and upward radiance $I_\lambda^\uparrow$, which allows to calculate $\alpha$ and retrieve $\tau$ and $r_{\text{eff,A}}$ at cloud top. The HALO Microwave Package (HAMP) enables to perform retrievals of

liquid water path $LWP$ and radar reflectivity $Z$ to discriminate between non-precipitating and precipitating cloud sections. Combining measured values of $I_\lambda^\uparrow$ by SMART and $LWP$ by HAMP, alternative values of $r_{\text{eff}}$ are retrieved, which are less influenced by sun-glint and 3D cloud radiative effects. Cloud top height is determined by the Water Vapour Lidar Experiment in Space (WALES) while the cloud base height is estimated from dropsondes or radar data.

In case of shallow trade wind cumulus, the heterogeneity of the cloud field has to be considered. This is especially important

at the average flight speed of HALO ($\approx 220\,\text{m s}^{-1}$) and different instrument field-of-views (FOV), being in the range of individual clouds. This is indicated by the high occurrence (63 %) of clouds with a horizontal size smaller than 300 m. To adapt the calculation of $N$ to this heterogeneous cloud field, a careful cloud masking and filtering for homogeneous cloud sections is crucial. Using cloud flagging and masking to find clouds, which cover the entire FOV, $N$ can be calculated for approximately 55 % of all measured clouds.

Three different methods to retrieve $N$ are presented. Possible biases from sun-glint in the solar wavelength range are avoided by using $LWP$ instead of $\tau$. Replacing $\tau$ by $LWP$, possible retrieval biases in method A resulting from sun-glint in the solar wavelength range measured by SMART, are avoided. Retrievals of $LWP$ from HAMP apply wavelengths with several micrometer (20 to 100 GHz), which are not influenced by atmospheric constituents, like aerosol particles. Using the radiance ratio retrieval of SMART to derive $r_{\text{eff}}$ from two infrared wavelengths, the remaining sun-glint influence on $r_{\text{eff}}$ and possi-

ble absolute calibration errors are reduced. Calculating the true cloud geometric thickness $\text{d}z$ from combined measurements of HAMP, WALES, and dropsondes the change of $LWC$ with height of the cloud profile is determined and is used as a correction factor, replacing the adiabatic assumption in the common retrieval. Determination of $LWP$ from HAMP further





allows to separate clouds for binned $LWP$ and to untangle the effects of varying $LWP$ on $\alpha$ which is a substantial advantage (McComiskey and Feingold, 2008). Radar reflectivity $Z$ provided by HAMP allows to discriminate between precipitating and non-precipitating clouds.

The measurements of $LWP$ must be performed within an uncertainty range of $\pm 30\,\mathrm{g\,m^{-2}}$ or better to achieve the $N$ accuracy

of $\pm 14\mathrm{cm^{-3}}$ from SMART. While $\pm 30\,\mathrm{g\,m^{-2}}$ is chosen as a conservative value of $LWP$ uncertainty from HAMP, a refined uncertainty estimation is ongoing, which will provide the the uncertainty of $LWP$ as a function of $LWP$. For clouds with higher geometric thickness, measured $LWP$ becomes more representative as information of the entire cloud is used instead of $\tau$ and $r_{\mathrm{eff}}$ from SMART representing the cloud-top only.

To bypass the adiabatic assumption in the calculation of $N$ the possibility to determine the adiabaticity from combined mi-

crowave radiometer measurements of $LWP$, lidar cloud top height $h_{\mathrm{top}}$ measurements and cloud base height $h_{\mathrm{base}}$ assumptions from dropsondes is investigated. Despite constant thermodynamic conditions in the flight area, concerning temperature and pressure, the distribution of humidity is heterogeneous leading to a variable $h_{\mathrm{base}}$. The cloud base height variability of several tens of meters does not allow to use a fixed $h_{\mathrm{base}}$ estimation from calculated lifting condensation level basing on dropsondes to obtain reasonable $N$.

Alternatively, cloud geometric thickness $\mathrm{d}z$ can be determined by the radar, despite it is limited to clouds with particles large enough to produce a detectable radar echo. Cloud base height depends on the selected threshold of $Z$ which causes an uncertainty $h_{\mathrm{base}}$ in the range of tens of meters. If cloud boundary determination could be achieved in the range of the smallest vertical resolution of the radar with $\pm 30\,\mathrm{m}$ and measurement with $\overline{LWP} = 100\,\mathrm{g\,m^{-2}} \pm 30\,\mathrm{g\,m^{-2}}$ are assumed, this results in $\Delta f_{\mathrm{calc}} = \pm 0.1$ which is better than the approach with $\mathrm{d}z$ from dropsonde estimations. Nevertheless, a resolution of $\pm 15\,\mathrm{m}$ is

only achieved in the optimal case for well-defined cloud edges. Cloud boundaries and thickness determined by radar are more precise but restricted to clouds with well-defined cloud edges, non-precipitating clouds and clouds with droplets detectable by the radar. When the cloud top height and cloud base height can be determined within an uncertainty of $\pm 25\,\mathrm{m}$ by the radar, the uncertainty of the adiabaticity is below $\pm 0.12$ but increasing for clouds with geometric thickness below 500 m. If the cloud base and top heights can not be determined within $\pm 25\,\mathrm{m}$ the adiabatic assumption is more accurate and should be applied.

The application of the three presented methods is shown for synthetic measurements of six different clouds with $N_{\mathrm{cld}}$ of $50\,\mathrm{cm^{-3}}$, $100\,\mathrm{cm^{-3}}$, and $200\,\mathrm{cm^{-3}}$ each following an adiabatic and sub-adiabatic cloud profile. Overall, the sensitivity study leads to the conclusion, that an appropriate retrieval of $r_{\mathrm{eff}}$ is the most important factor for the calculation of $N$. Special care must be taken by considering the penetration depth and the wavelength selection. Depending on $\tau$ or $LWP$ and $N$ the penetration depth can reach several tenth meters inside the cloud, not representing $r_{\mathrm{eff}}$ at cloud top. Uncertainty of $N$ has

to be considered carefully. The synthetic measurements and error analysis clearly indicate that uncertainty in $N$ results from underestimation of $r_{\mathrm{eff}}$ due to the penetration depth of the incoming solar radiation into the cloud which depends on $\tau$, $N$, and $LWP$. Therefore, clouds with low $\tau$ and $LWP$ are most effected by errors of $N$, reducing for clouds with increasing $\tau$. Replacing $\tau$ by the $LWP$ from independent measurements by passive microwave radiometers can improve the accuracy of estimated $N$. Retrieved $N$ of clouds with low $N$ and $\tau$, like trade wind cumulus, are more effected by errors compared to clouds

with higher $N$ and $\tau$. Considering only adiabatic clouds (I,III,V) this can be clearly assigned to errors in retrieved $r_{\mathrm{eff}}$. For





the sub-adiabatic cloud cases (II,IV,VI) the error in $N$ also includes the shortcoming of assuming an adiabatic cloud profile. Therefore, a further improvement is achieved, when the actual sub-adibatic profiles of the clouds (II,IV,VI) are considered. By determining the cloud geometry from active radar and lidar measurements and dropsondes, the resulting $\Gamma_{\mathrm{calc}}$ can be determined and used as a correction factor in the calculation of $N$. The sensitivity study shows that the differences between

modeled $N_{\mathrm{cld}}$ and retrieved $N_{\mathrm{C,lib}}$ or $N_{\mathrm{C,R}}$ with method C, are significantly reduced comparing to method A or B, for all three cloud cases. This indicates that a correction with $\Gamma_{\mathrm{calc}}$ is vital and necessary for the calculation of $N$ of shallow trade wind cumulus using remote sensing techniques. Otherwise systematic overestimation of retrieved $N$ is present and not feasible. Nevertheless, errors resulting from the cloud boundary determination form the radar and dropsondes will lead to uncertainties in the calculated lapse rate.

All three methods are applied to a homogeneous and a heterogeneous cloud section. Determination of $\mathrm{d}z$ was relatively uncertain and method C was excluded from the statistical analysis. Probability density functions of $LWP$, $r_{\mathrm{eff}}$, and $N$ of the two cloud cases are presented in Sec. 6. Correlations of $\mathcal{R}_{532}$ to $N_{\mathrm{B}}$ for two binned $LWP_{\mathrm{B}}$ are shown. These can be used to validate modeled $\mathcal{R}_{532}$, to describe the sensitivity of $\mathcal{R}_{532}$ with respect to $N$, and allow to parameterize the Twomey effect better. Further testing of the proposed method to longer flight sections with more homogeneous cloud fields is necessary, to in-

crease the covered natural variability of trade-wind cumulus. Despite remaining uncertainties and assumptions, the application of $\Gamma_{\mathrm{calc}}$, the separation for different $LWP$, and the smaller FOV of all instruments, allow to investigate the cloud-radiation interactions better compared to large-scale averaging satellite measurements.

*Acknowledgements.* This research was funded by the German Research Foundation (DFG, HALO-SPP 1294). The authors acknowledge the support by the Deutsche Forschungsgemeinschaft (DFG) through grants CR 111/10-1, PF 384/7-1/2, PF 384/16-1, and WE 1900/35-1.

Special thanks also go to Felix Ament and Heike Konow which provide the calibrated HAMP radar and microwave data. Additionally, the authors thank the pilots and appreciate the support by the Flugbereitschaft of the German Aerospace Center (DLR) and enviscope GmbH for preparation and testing of SMART.





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
