# Peer review of "Improvement of Airborne Retrievals of Cloud Droplet Number Concentration of Trade Wind Cumulus Using a Synergetic Approach"

_Atmospheric Measurement Techniques, 2018_

## Referee Comment (RC1) · Anonymous Referee #2 · 21 Jan 2019

Wolf et al. provide an interesting mutli-sensor approach towards better constraining CDNC estimates in trade wind cumuli. These clouds present a challenge to remote sensing of their microphysical parameters, especially in the Vis/NIR portion of the spectrum. Using a combined V/NIR and active/passive microwave measurements, while in and of itself is not novel, the use of MW LWP measurements to help quantify the degree of adiabaticity in the cloud retrieval is. It definitely contributes to the existing body of literature. I recommend that the manuscript be published subject to a few relatively minor revisions.

Specific comments:

[Figure]

P2 L32 : Perhaps mention that Q_ext is around 2 and thus the coefficient in equation 1 is 2/3. It would be helpful for those not as familiar with VNIR cloud retrievals.

P2 L18: Bennartz and Rausch (2017) doesn't assume a constant LWC vertically, but a sub-adiabatically stratified, linearly increasing LWC of roughly 80% of the purely adiabatic value

P5L25: The k-parameter shows up in equation 3, but there is no mention of what k represents until page 12. It may be helpful to provide the reader a little more information on k rather than leaving them hanging for 7 pages.

P11L19: With regard to the effective radius retrievals, SMART's absorption channel around 1.6 microns, which has a significant amount of vertical penetration into the cloud relative to 3.7 or 2.1 micron absorption channels. For an adiabatically stratified cloud, the re represents the cloud-top value. So, 1.6 microns would underestimate the true re,LWP and thus N. I understand that it is a limitation of the instrument, but it may be worth mentioning this and how it may impact your retrievals especially when comparing to microwave LWP. It is mentioned in the conclusions on P31 of the manuscript, but would be worth mentioning again in this section.

P19L7: The study used radar measurements to identify potentially precipitating observations. Since Z is more sensitive to larger droplets, it can't easily identify drizzle cases, as you mention. For the cases in section 6, I think it may be helpful to augment the radar with a VNIR ratio of cloud geometrical thickness and CDNC to identify potentially drizzling cases that radar can't identify. Van Zanten and Stevens (2005) for example establishes ratios of $H^3/N$ for identification of drizzle in stratocumuli. For the transition to trade cumuli, this may not be clear-cut, but nevertheless is may help reduce the misclassification of drizzling clouds, which would affect the statistics on retrieved optical parameters.

P25: Figure 6. I don't see any mention of it in the body of the manuscript.

P32: Of the three methods A,B, & C, which is best? and when? I didn't feel like I got a clear and concise message on that in the conclusions. I feel like the conclusion section broadly covered this, but not concisely.

---

## Referee Comment (RC2) · Anonymous Referee #3 · 7 Feb 2019

The authors discussed methods using various wavelengths and passive/active sensors from the aircraft measurement to better retrieve cloud droplet number concentration. Past studies on this topic are adequately cited and materials are treated quantitatively. Overall, the sentences are well written. The reviewer suggests several minor points before recommending publication.

In Fig.1 b), why do curves cross each other? The reviewer guesses that these curves would be simple and not touch each other.

The reviewer feels that '7. Conclusions' is a bit lengthy. Hopefully, this section can be shorter, focusing on the main outcome.

[Figure]

All through the manuscript, I found grammatical mistakes and typographical errors frequent ( just one sample, P28L18 eg.g.,). Proofreading by a professional editor will make this manuscript much better.
* * *

---

## Author Comment (AC1) · 26 Feb 2019

**Response to Anonymous Referee #2**

The comments of the reviewer have been helpful to improve the manuscript. We thank the reviewer for the suggestions. Detailed replies on the reviewers comments are given below. The reviewers comments are given bold while our replies are written in regular roman letters. Citations from the revised manuscript are given as indented and italic text.

**P2 L32 : Perhaps mention that Q_ext is around 2 and thus the coefficient in equation 1 is 2/3. It would be helpful for those not as familiar with VNIR cloud retrievals.**
A: Now the derivation of the equation is described in more detail. Explicitly stating Qext approx. 2. The change in the manuscript is copied here as a screenshot to provide better readability because of the equations.

20 Assuming an adiabatic cloud, the $LWC$ increases linearly with height and the liquid water path $LWP$ is determined by integrating over the altitude $z$ from cloud base (CB) to cloud top (CT):

$$LWP = \int_{CB}^{CT} LWC(z)\,dz = \frac{4}{3}\cdot\pi\cdot\rho_w \cdot \int_{CB}^{CT} N(z)\cdot r_3^3(z)\,dz \tag{1}$$

with the density of liquid water $\rho_w$, the cloud droplet number concentration $N(z)$ in height $z$, and the mean volumetric radius $r_3$. Following Hansen and Travis (1974) and Stephens (1978) the cloud optical thickness $\tau$ is related to the $LWP$ by:

$$\quad \tau = \int_{CB}^{CT} \sigma_{ext}\,dz = \int_{CB}^{CT} \pi \int_0^\infty Q_{ext}(x)\cdot N(r,z)\cdot r^2\,dr\,dz = \int_{CB}^{CT} \pi \cdot Q_{ext}(\overline{x})\cdot N(z)\cdot r_2^2\,dz \tag{2}$$

with the extinction coefficient $\sigma_{ext}$, the extinction efficiency factor $Q_{ext}$ which is approximately 2 for cloud droplets in the solar wavelength range, the size parameter $x = (2\cdot\pi\cdot r)/\lambda$, and the mean radius $r_2$. According to Martin et al. (1994) the effective radius $r_{eff}$ correlates with the mean surface radius $r_2$ and the mean volume radius $r_3$ of the droplet size distribution given by:

$$k = \left(\frac{r_3}{r_{eff}}\right)^3 = \left(\frac{r_2^3}{r_3^2}\right)^6. \tag{3}$$

This relation depends on the shape of the droplet size distribution and is referred as the $k$-parameter in the literature. Using $k$ as the distribution shape factor, $r_2$ and $r_3$ in Eq. 1 and Eq. 2 are replaced by $r_{eff}$ leading to:

$$\tau = \frac{3\cdot\int_{h_{CB}}^{h_{CT}} LWC(z)\cdot dz}{2\cdot\rho_w\cdot r_{eff}}. \tag{4}$$

**P2 L18: Bennartz and Rausch (2017) doesn't assume a constant LWC vertically, but a sub-adiabatically stratified, linearly increasing LWC of roughly 80% of the purely adiabatic value**
A: Bennartz and Rausch (2017) are extracted from the enumeration and are named separately with the correct cloud profile description.

*"… They are a useful tool, providing large spatial and temporal data sets. Based on passive remote sensing in the solar and terrestrial wavelength range, N is estimated combining the results of bi-spectral retrievals of cloud optical thickness and reff and cloud top temperature TCT by Brenguier et al. (2000), Quaas et al. (2006), and Zeng et al. (2014). They assume a constant LWC and N throughout the cloud vertical profile, which is not necessarily fulfilled in nature. Slightly deviating, Bennartz and Rausch (2017) assume a sub-adiabatic profile where the LWC increases linearly with height by approx. 80% with respect to the adiabatic value."*

**P5L25: The k-parameter shows up in equation 3, but there is no mention of what k represents until page 12. It may be helpful to provide the reader a little more information on k rather than leaving them hanging for 7 pages.**
A: The k-parameter is introduced early in the section of the text which was suggested by the reviewer. The description of the k-parameter is included in the derivation of the tau-lwc-relation. Please see answer to comment #1.

**P11L19: With regard to the effective radius retrievals, SMART's absorption channel around 1.6 microns, which has a significant amount of vertical penetration into the cloud relative to 3.7 or 2.1 micron absorption channels. For an adiabatically stratified cloud, the re represents the cloud-top value. So, 1.6 microns would underestimate the true re,LWP and thus N. I understand that it is a limitation of the instrument, but it may be worth mentioning this and how it may impact your retrievals especially when comparing to microwave LWP. It is mentioned in the conclusions on P31 of the manuscript, but would be worth mentioning again in this section.**
A: Now the potential bias in the retrieved reff and according estimated N due to the varying penetration depth of the reflected solar radiation is stated in this section.

*"The effective radius is derived with the radiance ratio method, using a ratio of measurements at 1050 nm and 1645 nm. Compared to retrievals using larger wavelength, e.g. 2.1 or 3.7μm, reff retrieved by the SMART measurements does not only represent the cloud particles at cloud top. The vertical weighting function for 1.6 \mu m covers significant amount of information from lower cloud layers (Platnick, 2000). Therefore, retrieved reff are smaller than the actual cloud droplet size at CT which are considered in Eq. 12 to calculate N. This leads to a systematic overestimation of N calculated from SMART measurements."*

**P19L7: The study used radar measurements to identify potentially precipitating observations. Since Z is more sensitive to larger droplets, it can't easily identify drizzle cases, as you mention. For the cases in section 6, I think it may be helpful to augment the radar with a VNIR ratio of cloud geometrical thickness and CDNC to identify potentially drizzling cases that radar can't identify. Van Zanten and Stevens (2005) for example establishes ratios of $H^3$/N for identification of drizzle in stratocumuli. For the transition to trade cumuli, this may not be clear-cut, but nevertheless is may help reduce the misclassification of drizzling clouds, which would affect the statistics on retrieved optical parameters.**
A: The publications by Van Zanten et al. (2005) and Pawlowska et al. (2003) are mentioned in the text. The correlation of $dz^3$ on N, as a measure for the drizzle reduction rate, was tested on the data presented in the manuscript but did not show any statistical significant separation for

drizzle and non-drizzle sections. As mentioned in the text the retrieval of N will be biased in case of drizzle events and therefore, a separation on basis of retrieved N and the measured cloud geometric thickness is not possible. Despite that, a plot of dz³ as a function of N is provided here:

[Figure]

"*Estimation of the drizzle rate on basis of dz and N as proposed by Pawlowska and Brenguier (2003) and vanZanten et al. (2005) is not possible as retrieved N is biased by the process of drizzle formation and, therefore, not applicable with the presented instrument setup of HALO.*"

**P25: Figure 6. I don't see any mention of it in the body of the manuscript.**
A: Figure 6 is now discussed in the text. Mean and median values for the analysis are included and summarized in a table.
"*Figures 6a and b 5 show the normalized probability density function (PDF) of LWP retrieved by HAMP and SMART separated for precipitating and non-precipitating clouds. For the non-precipitating clouds, the distributions of LWP retrieved by SMART and HAMP are dominated by clouds below 100 gm. Higher LWP are obtained for regions with precipitation, where the distribution is shifted towards larger values of LWP. The PDF of LWP_A and LWP_B show a dominant mode at around 150 gm-2. A second smaller mode is present for LWP_A at 80 gm-2 and LWP_B at 50 gm-2 for both instruments. The agreement of the LWP retrievals, utilizing reflected solar radiation from CT (method A) and passive microwave measurements (method B), indicate that the cloud microphysical properties are sufficiently determined by the SMART retrieval, despite the assumption of an adiabatic cloud profile in method A.*"

**P32: Of the three methods A,B, & C, which is best? and when? I didn't feel like I got a clear and concise message on that in the conclusions. I feel like the conclusion section broadly covered this, but not concisely.**
A: The author tried to formulate the conclusion more precisely and emphasizing the advantage of each method under the specific cases. Simultaneously the length of the conclusion was reduced.

*"From the synthetic measurements and the two cloud cases it can be concluded that method A is suggested for optically thin clouds with (LWP < 100 gm3) while method B should be preferred for optically thicker clouds. For homogeneous clouds when the cloud boundaries can be determined precisely from the active radar, lidar, and dropsonde measurements, the resulting gamma_calc can be determined and used as a correction factor in the calculation of N as the optimal case. The synthetic measurements showed that the differences between modeled Ncld and retrieved NC;lib or NC;R with method C, are significantly reduced comparing to method A or B, for all three cloud cases. This indicates that a correction with gamma_calc is vital and necessary for the calculation of N of shallow trade wind cumulus using remote sensing techniques. Otherwise systematic overestimation of retrieved N is present and not feasible."*

Please also see the latex difference file, where the changes become visible in the manuscript.

---

## Author Comment (AC2) · 26 Feb 2019

**Response to Anonymous Referee #3**

The comments of the reviewer have been helpful to improve the manuscript. We thank the reviewer for the suggestions. Detailed replies on the reviewers comments are given below. The reviewers comments are given bold while our replies are written in regular roman letters. Citations from the revised manuscript are given as indented and italic text.

**In Fig.1 b), why do curves cross each other? The reviewer guesses that these curves would be simple and not touch each other.**
A: Due to low cloud optical thickness, and liquid water path / liquid water content, the calculated albedo / sensitivity zeta are easily affected by small effective radius and according dependence on the phase function on reff within the simulations. Therefore the simulations have been repeated with higher precision, reducing the noise and making the plot in Fig. 1b) more clear. Despite that, some crossing of the lines is still present for higher DNC, which could not be removed completely.

[Figure]

**The reviewer feels that '7. Conclusions' is a bit lengthy. Hopefully, this section can be shorter, focusing on the main outcome.**
A: The author tried to (re-)phrase the statements more clear and precise by simultaneously reducing the length of the section. Parts of the former conclusion where transfereed to the uncertainty estimation and sensitivity.

*"From the synthetic measurements and the two cloud cases it can be concluded that method A is suggested for optically thin clouds with (LWP < 100 gm-3) while method B should be preferred for optically thicker clouds. For homogeneous clouds when the cloud boundaries can be determined precisely from the active radar, lidar, and dropsonde measurements, the resulting gamma_calc can be determined and used as a correction factor in the calculation of N as the optimal case. The synthetic measurements showed that the differences between modeled Ncld and retrieved NC;lib or NC;R with method C, are significantly reduced comparing to method A or*

*B, for all three cloud cases. This indicates that a correction with gamma_calc is vital and necessary for the calculation of N of shallow trade wind cumulus using remote sensing techniques. Otherwise systematic overestimation of retrieved N is present and not feasible."*

**All through the manuscript, I found grammatical mistakes and typographical errors frequent ( just one sample, P28L18 eg.g.,). Proofreading by a professional editor will make this manuscript much better.**
A: Careful proofreading has been performed to minimize typos.

Please also see the latex difference file, where the changes become visible in the manuscript.

---

## Author Response (AR1)

[revised manuscript text omitted]
_{\text{ad}}}\right)^2 (\Delta f_{\text{ad}})^2 + \left(\frac{\partial N}{\partial \Gamma_{\text{add}}}\right)^2 (\Delta \Gamma_{\text{add}})^2 + \left(\frac{\partial N}{\partial \tau}\right)^2 (\Delta \tau)^2 + \left(\frac{\partial N}{\partial r_{\text{eff}}}\right)^2 (\Delta r_{\text{eff}})^2} \tag{16}$$

and analogous for Eq. (12) and Eq. (14). All uncertainties of $N$ presented in the following sections are based on calculation by this approach. The uncertainties of the single parameters assumed in the calculations are summarized below.

For method A, B, and C, the uncertainty of $k$, representing the shape of the droplet size distribution, is set to $k = 0.8 \pm 0.1$ according to the range of values suggested by Martin et al. (1994) and Pontikis and Hicks (1992).

For methods A and B the degree of adiabaticity $f_{\text{ad}}$ is fixed to one. In that case, no uncertainty in a measurement scene is attributed to $f_{\text{ad}}$. For method C, the uncertainty of $f_{\text{calc}}$ is determined by the uncertainty of  $h_{\text{CT}}$, $h_{\text{CB}}$, and retrieved $LWP$ following (Eq. (13). Cloud top height from WALES is determined with an accuracy of  $\Delta h_{\text{CT}} = \pm 20\,\text{m}$. The cloud base height is derived from single dropsondes and, therefore, prone to horizontal variability of $T$, $p$, and $T_{\text{d}}$. Based on an analysis of different dropsondes in close vicinity,  a cloud base height $h_{\text{LCL}} = 660\,\text{m} \pm 35\,\text{m}$ is assumed. The evaluation of all dropsondes show that the thermodynamic conditions in the selected area stayed constant ($\Delta T < 2\,\text{K}$ and $\Delta p < 4\,\text{hPa}$) during the flight time with  $h_{\text{CT}} \approx 1800\,\text{m}$,  $T_{\text{CT}} = 20.2°\text{C}$, and  $p_{\text{CT}} = 820\,\text{hPa}$. The accuracy of the deployed Vaisala dropsondes RD94 is reported to be within $\Delta T = \pm 0.2\,\text{K}$ and $\Delta p = \pm 0.4\,\text{hPa}$. Uncertainties of $N_C$ caused by errors in $\Gamma_{\text{ad}}$ are, therefore, negligible compared to the influence of $\tau$ and $r_{\text{eff}}$.

The adiabatic increase of $LWC$ with height calculated from the Clausius-Clapeyron-Equation depends mostly on cloud top temperature  $T_{\text{CT}}$ and to a lower degree on cloud top pressure  $p_{\text{CT}}$. Therefore, $\Gamma_{\text{ad}}$ depends on  $T_{\text{CT}}$ and $p_{\text{CT}}$, too. The cloud droplet number concentration is mostly effected by the assumed  $T_{\text{CT}}$ whereby $p_{\text{CT}}$ is only of minor contribution. Despite that, the cloud top pressure more strongly affects warm than cold clouds (Grosvenor et al., 2018b). For the uncertainty calculation, a temperature difference of 2 K is considered, which changes $\Gamma_{\text{ad}}$ by $\pm 0.1 \cdot 10^{-3}\,\text{g}\,\text{m}^{-3}\,\text{m}^{-1}$ for the reference value of $2.5 \cdot 10^{-3}\,\text{g}\,\text{m}^{-3}\,\text{
[revised manuscript text omitted]
_{\text{med,S}}$ | $\overline{r}_{\text{eff,A}}$ | $r_{\text{eff,med,A}}$ | $\overline{h}_{\text{ct,L}}$ | $\overline{N}_A$ | $\overline{N}_{A,\text{med}}$ | $\overline{N}_B$ | $\overline{N}_{B,\text{med}}$ | $\overline{LWP}_A$ | $\overline{LWP}_B$ |
|---|---|---|---|---|---|---|---|---|---|---|---|
| np | 3.5 | 2.2 | $23.2\,\mu\text{m}$ | $21.1\,\mu\text{m}$ | 1798 m | $17\,\text{cm}^{-3}$ | $14\,\text{cm}^{-3}$ | $25\,\text{cm}^{-3}$ | $12\,\text{cm}^{-3}$ | $72\,\text{g\,m}^{-3}$ | $82\,\text{g\,m}^{-3}$ |
| p | 11.3 | 7.0 | $25.1\,\mu\text{m}$ | $24.5\,\mu\text{m}$ | 1988 m | $47\,\text{cm}^{-3}$ | $17\,\text{cm}^{-3}$ | $53\,\text{cm}^{-3}$ | $25\,\text{cm}^{-3}$ | $170\,\text{g\,m}^{-3}$ | $203\,\text{g\,m}^{-3}$ |

[revised manuscript text omitted]